# ALIGNMENT AND OUTER SHELL ISOTROPY FOR HYPERBOLIC GRAPH CONTRASTIVE LEARNING

## ABSTRACT

Learning good self-supervised graph representations that are beneficial to downstream tasks is challenging. Among a variety of methods, contrastive learning enjoys competitive performance. The embeddings of contrastive learning are arranged on a hypersphere that enables the Cosine distance measurement in the Euclidean space. However, the underlying structure of many domains such as graphs exhibits highly non-Euclidean latent geometry. To this end, we propose a novel contrastive learning framework to learn high-quality graph embedding. Specifically, we design the alignment metric that effectively captures the hierarchical data-invariant information, as well as we propose a substitute of uniformity metric to prevent the so-called dimensional collapse. We show that in the hyperbolic space one has to address the leaf- and height-level uniformity which are related to properties of trees, whereas in the ambient space of the hyperbolic manifold, these notions translate into imposing an isotropic ring density towards boundaries of Poincaré ball. This ring density can be easily imposed by promoting the isotropic feature distribution on the tangent space of manifold. In the experiments, we demonstrate the efficacy of our proposed method across different hyperbolic graph embedding techniques in both supervised and self-supervised learning settings.

## 1 INTRODUCTION

Learning features in hyperbolic spaces has drawn a lot of interest (Bronstein et al., 2017; Sun et al., 2021). Hyperbolic spaces are characterized by their negative curvature, where the distance between two points can grow exponentially in contrast to the Euclidean space where the distance grows linearly. Compared with the Euclidean space, hyperbolic spaces have several advantages: 1) Geodesic distance measure, 2) Better representation of hierarchical structures, 3) Increased capacity, and 4) Improved generalization.

Contrastive learning pulls related samples closer in the Euclidean space while repelling unrelated samples from each other (Chen et al., 2020a;b; Gao et al., 2021; Zhang et al., 2022; Wang & Isola, 2020). Despite promising results of contrastive learning (CL) (Chen et al., 2020a; Zhang et al., 2022; Gao et al., 2021), CL suffers from a fundamental limitation in the ability to model complex patterns as CL inherently bounded by the dimensionality of the embedding space (Nickel & Kiela, 2017). In typical CL, embeddings are arranged on a hypersphere and compared by the inner (dot) product or the Cosine distance (Figure 1). However, the underlying structure in many domains such as graph data is inherently hierarchical (Bronstein et al., 2017). The quality of the representations is determined by how well the geometry of the embedding space matches the structure of the data, and has been explored in various works (Mathieu et al., 2019).

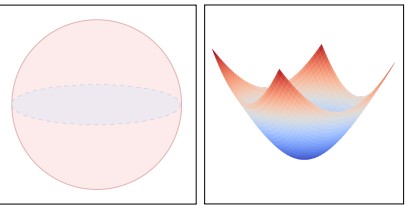

Figure 1: Hypersphere *vs*. Hyperbola.

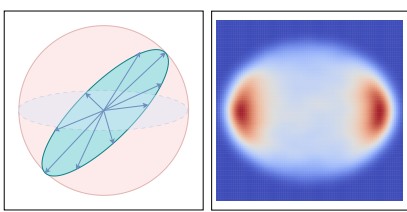

Figure 2: Dimensional Collapse on (left) the $\ell_2$ ball *vs*. (right) the Poincaré disk).

Wang & Isola (2020) showed that the quality of representation produced via contrastive learning is characterized by two key factors: alignment and uniformity Wang & Isola (2020). Good alignment ensures the preservation of distinct information contained by samples, while excessive alignment

may lead to dimensional collapse where features become overly concentrated on specific points or subspace (Figure 2) (Jing et al., 2021; Grill et al., 2020). To counteract this effect, the so-called uniformity constraints are typically imposed to encourage samples to span the entire space evenly to increase the feature diversity. In the Euclidean space, maximizing the pairwise distance between samples on the hypersphere helps achieve a roughly uniform distribution on the surface.

However, since the hyperbolic space is non-compact and has infinite volume, maximizing the uniformity directly in the hyperbolic space leads to pushing all samples towards the boundary of Poincaré ball which has infinite volume. Thus, uniformity cannot be achieved. Therefore, alignment with/without uniformity results in the Hyperbolic dimensional collapse (HDC), which can be characterized by decline of Effective Rank (Roy & Vetterli, 2007) of features in the ambient space of the hyperbolic manifold[1]. The HDC is illustrated in Figures 4a, 4c & 4d.

In this paper, we investigate the phenomenon of HDC. We adopt the Poincaré model and introduce a novel framework called Hyperbolic Graph Contrastive Learning (HyperGCL). The primary goal of HyperGCL is to generate high-quality graph embeddings that avoid the HDC and can be effectively utilized in various downstream tasks. To this end, we revisit the concept of alignment and uniformity for the Hyperbolic embeddings. We adopt the Hyperbolic distance[2] to measure the alignment in the hyperbolic space.

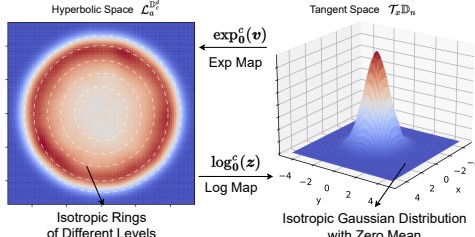

Figure 3: Mapping isotropic Gaussian from the tangent plane at $\mathbf{0}$ to the ambient space of the hyperbolic manifold (and back).

> As the notion of uniformity is undefined for manifolds other than hypersphere, especially for a manifold with an infinite volume of hyper-surface, instead of talking about the "uniformity" of hyperbolic space, we will talk about the level-wise isotropic rings in the ambient space of the hyperbolic manifold (Figure 3 (left)). We are interested in imposing high density of features uniformly distributed along the ring circumference close to the boundary of Poincaré ball. We call it the outer isotropic shell or simply put, isotropic shell. To optimize it, we discover that enforcing an isotropic Gaussian distribution on the tangent plane at $\mathbf{0}$ to the Poincaré ball results in such an isotropic shell, as shown in Figure 3. We are interested in such an ambient space because it corresponds to a deep tree with uniformly distributed leaves (Fig. 4b).

Below we summarize our contributions as follows:

i. We investigate the dimension collapse problem in the hyperbolic contrastive learning, and we associate the dimensional collapse with the tree "leaf collapse" and "height collapse", describing it from the point of view that trees can be represented in hyperbolic spaces (Ganea et al., 2018).

ii. We propose a novel graph contrastive learning framework in the hyperbolic space to generate high-quality graph embedding for the various downstream tasks.

iii. To alleviate the dimensional collapse problem, identified and measured as decline in Efficient Rank of features in the ambient space of the hyperbolic manifold, we propose a new isotropic Gaussian loss operating on the tangent space of manifold at $\mathbf{0}$ which forces the desired feature distribution (isotropic shell) in the ambient space of the hyperbolic manifold. We show that imposing the isotropic Gaussian loss on the tangent space increases the Effective Rank of feature representations on the tangent plane, and we show that the Effective Rank measured in the ambient space correlates with the effective rank measured on the tangent plane.

## 2 PRELIMINARIES

### 2.1 ALIGNMENT AND UNIFORMITY ON HYPERSPHERE $\mathcal{S}^{m-1}$

Contrastive learning is commonly used to encourage learned feature representations for positive pairs to be similar while pushing away features from randomly sampled negative pairs. It is known

---

[1]In simple terms, the ambient space of the Hyperbolic manifold is the space in which hyperbola is defined.

[2]Hyperbolic distances are known to approximate tree/hierarchical distances (Theorem 1).

that representations should capture the information shared between positive pairs while remaining invariant to nuisance/noise factors (Tschannen et al., 2019; Wu et al., 2018). When the representation resides on the hypersphere $\mathcal{S}^{m-1}$ ($l_2$ normalized), Wang & Isola (2020) argue that the above properties can be achieved through optimization of the following objective:

$$\mathcal{L}_{contrastive} = \underbrace{\mathop{\mathbb{E}}_{(\boldsymbol{x},\boldsymbol{y})\sim p_{\mathrm{pos}}} \|f(\boldsymbol{x}) - f(\boldsymbol{y})\|_2^2}_{\mathcal{L}_A^{\mathbb{R}^d}} + \underbrace{\log \mathop{\mathbb{E}}_{\substack{\mathrm{i.i.d} \\ \boldsymbol{x},\boldsymbol{y}\sim p_{\mathrm{data}}}} \left[ e^{-t\|f(\boldsymbol{x})-f(\boldsymbol{y})\|_2^2} \right]}_{\mathcal{L}_U^{\mathbb{R}^d}}, \tag{1}$$

where $f(\cdot)$ denotes the encoder with the $l_2$-normalized output, $i.e.$, $\|f(x)\|_2 = 1$. One can observe that (i) **Alignment** $\mathcal{L}_A^{\mathbb{R}^d}$ makes two samples of a positive pair to be mapped to nearby feature vectors, and thus be (mostly) invariant to undesired nuisance/noise factors, whereas (ii) **Uniformity** $\mathcal{L}_U^{\mathbb{R}^d}$ forces feature vectors be roughly uniformly distributed on the unit ball $\mathcal{S}^{m-1}$, diversifying features.

**Dimensional Collapse.** Often referred to as spectral collapse (Liu et al., 2019a), dimensional collapse is common in representation learning as shown in Figure 2. It occurs when the embedding space is dominated by a small number of large singular values, while the remaining singular values decay rapidly as the training progresses. This phenomenon limits the representation power of high-dimensional spaces by restricting the diversity of information that can be learned. In the framework of optimizing the alignment and uniformity, optimizing the uniformity helps alleviate the dimensional collapse by encouraging features to be uniformly distributed in the entire latent space.

## 2.2 HYPERBOLIC GEOMETRY

To describe our HyperGCL, we begin with a brief review of the hyperbolic geometry and its several properties that will be used in our model.

**Poincaré Ball Model.** A hyperbolic space $\mathbb{H}$ is a complete, connected Riemannian manifold with constant negative sectional curvature. Cannon et al. (1997) describe five common hyperbolic models. In this paper, we choose the Poincaré ball $\mathbb{D}_c^d := \left\{ \boldsymbol{p} \in \mathbb{R}^n \mid \|\boldsymbol{p}\|^2 < \frac{1}{c} \right\}$ as our basic model (Nickel & Kiela, 2017; Tifrea et al., 2018), where $\frac{1}{c} > 0$ is the radius of the ball. The Poincaré ball is coupled with a Riemannian metric $g_{\mathbb{D}}(\boldsymbol{p}) = \frac{4c}{(1-\|\boldsymbol{p}\|^2)^2} g_{\mathbb{E}}$, where $\boldsymbol{p} \in \mathbb{D}_c^d$ and $g_{\mathbb{E}}$ is the canonical metric of the Euclidean space. The hyperbolic space is globally differomorphic to the Euclidean space.

**Definition 1 (Riemannian distance in $\mathbb{D}_c^d$)** *For $\boldsymbol{p}, \boldsymbol{q} \in \mathbb{D}_c^d$, the Riemannian distance on the Poincaré ball induced by its metric $g_{\mathbb{D}}$ is defined as $D_c(\boldsymbol{p}, \boldsymbol{q}) = \frac{2}{\sqrt{c}} \tanh^{-1}\left(\sqrt{c}\| - p \oplus q\|_2\right)$ where $\oplus$ is the Möbius addition and it is clearly differentiable.*

**Definition 2 (Tangent Space)** *The tangent space $\mathcal{T}_{\boldsymbol{x}}\mathbb{D}_c^n$ ($\boldsymbol{x} \in \mathbb{D}_c^n$) is defined as the first-order approximation of $\mathbb{D}_c^n$ around ponit $\boldsymbol{x}$ : $\mathcal{T}_{\boldsymbol{x}}\mathbb{D}_c^n := \left\{ \boldsymbol{v} \in \mathbb{R}^{n+1} : \langle \boldsymbol{v}, \boldsymbol{x} \rangle = 0 \right\}$.*

To perform operations in the hyperbolic space, the bijective map from $\mathbb{R}^n$ to $\mathbb{D}_c^d$ maps Euclidean vectors to the hyperbolic space. The so-called exponential map performs such a mapping (the logarithmic map performs the inverse mapping).

**Definition 3 (Exponential/Logarithmic Map)** *The exponential map $\exp_{\boldsymbol{x}}^c(\cdot)$ is a function from $T_{\boldsymbol{x}}\mathbb{D}_c^d \cong \mathbb{R}^n$ to $\mathbb{D}_c^d$. The logarithmic map $\log_{\boldsymbol{x}}^c(\cdot)$ maps from $\mathbb{D}_c^d$ to $T_{\boldsymbol{x}}\mathbb{D}_c^d$. These maps are defined as:*

$$\exp_{\boldsymbol{x}}^c(\boldsymbol{v}) = \boldsymbol{x} \oplus (\tanh(\frac{\sqrt{c}\lambda_{\boldsymbol{x}}^c}{2}\|\boldsymbol{v}\|)\frac{\boldsymbol{v}}{\sqrt{c}\|\boldsymbol{v}\|}) \text{ and } \log_{\boldsymbol{x}}^c(\boldsymbol{y}) = \frac{2}{\sqrt{c}\lambda_{\boldsymbol{x}}^c} \tanh^{-1}(\sqrt{c}\|-\boldsymbol{x} \oplus \boldsymbol{y}\|)\frac{-\boldsymbol{x} \oplus \boldsymbol{y}}{\|-\boldsymbol{x} \oplus \boldsymbol{y}\|},$$
$$\tag{2}$$

*where $\lambda_{\boldsymbol{x}}^c = \frac{2}{1-c\|\boldsymbol{x}\|^2}$ is the conformal factor that scales the local distances and $\|\cdot\|$ is the $\ell_2$ norm.*

We use $\exp_{\boldsymbol{0}}^c(\cdot)$ and $\log_{\boldsymbol{x}}^c(\cdot)$ to transition between the Euclidean and Poincaré ball representations.

**Tree Structure in Poincaré Ball.** Hierarchical relations between data points call for Hyperbolic embeddings. As the volume grows exponentially faster in the hyperbolic space ($e.g.$, towards

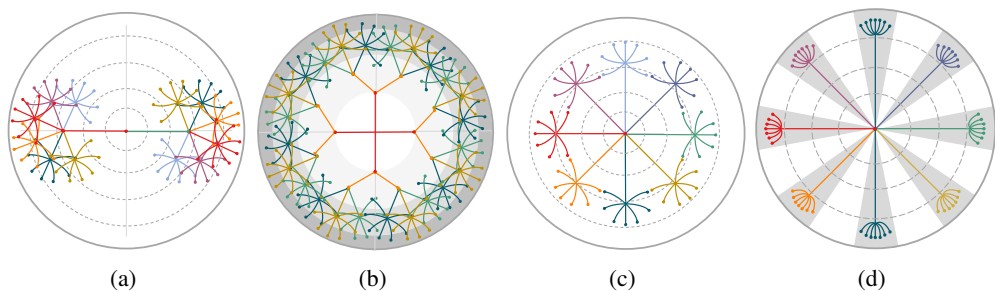

Figure 4: Trees embedded in Poincaré disk. Fig. 4a, 4c & 4d are tree embeddings under the dimensional collapse, whereas Fig. 4b is plausible "healthy" tree embedding (see text for details).

boundary of Poincaré Ball) than in the Euclidean space, the hyperbolic space is suitable for embedding hierarchical structures with constant branching factors and an exponential number of nodes. Such a property is formally discussed by Ganea et al. (2018) who state that any tree can be embedded into a Poincaré disk ($n = 2$) with low distortion (unlike embedding the tree into the Euclidean space (Linial et al., 1995)). Such a property is formally stated in the theorem below.

**Theorem 1** *(Ganea et al., 2018) Given a Poincaré ball $\mathbb{D}_c^d$ with an arbitrary dimension $n \geq 2$ and any set of points $p_1, \cdots, p_m \in \mathbb{D}_c^d$, there exists a finite weighted tree $(T, d_T)$ and an embedding $f : T \to \mathbb{D}_c^d$ such that for all $i, j$, $\left| d_T \left( f^{-1} \left( \boldsymbol{x}_i \right), f^{-1} \left( \boldsymbol{x}_j \right) \right) - d_{\mathbb{D}} \left( \boldsymbol{x}_i, \boldsymbol{x}_j \right) \right| = \mathcal{O}(\log(1 + \sqrt{2}) \log(m))$, where $d_T(\cdot, \cdot)$ and $d_{\mathbb{D}}(\cdot, \cdot)$ represent the tree and Hyperbolic distance, respectively.*

## 3 METHODOLOGY

**Dimensional Collapse in the Hyperbolic Space.** Preventing the dimensional collapse in representation learning is of utmost importance. Wang & Isola (2020) provide a new perspective on this issue by introducing a uniformity metric that limits the degree of collapse in representations. They demonstrate that in order to maintain diverse information for downstream tasks, the learned embeddings should be evenly distributed on the hypersphere $\mathcal{S}^{m-1}$. In the Euclidean space, a typical collapse mode occurs when features collapse to a single point or a subspace, as depicted in Figure 2 (left). However, in the hyperbolic space, the collapse mode (Figure 2 (right)) differs due to the tree property, as illustrated by Theorem 1.

Due to the exponential growth of hyperbolic space, direct application of uniformity fails to "fill" that infinite volume uniformly. The uniformity is the ambient space of the hyperbolic manifold is also meaningless as along boundaries of Poincaré ball, the hyperbolic space grows infinitely.

> The dimensional collapse in Hyperbolic space results in an unbalanced distribution in the ambient space of the hyperbolic manifold. Figure 4 shows the collapse mode may be associated with inadequately constructed tree embeddings. A data point near the center of the Poincaré ball is considered the root node, while data points near the boundary of the ball are leaf nodes. One possible collapse mode is referred by us to as "leaf collapse," where the embedding of the tree node collapses into several dense regions at a specific level of the tree. Figure 4a shows an example of this collapse, which is similar to the collapse observed on the hypersphere when the model maps points in the opposite direction of the ball. Another collapse mode, called by us as "height collapse" (Figure 4c), occurs when the underlying hierarchy results in a shallow tree, limiting discriminative hierarchical relationships. Figure 4d includes both "leaf collapse" and "height collapse" that limit the efficient use of the embedding space, and the expressive power of the hyperbolic space for downstream tasks. Figure 4b shows a healthy embedding. The grey rings illustrate density of feature vectors in the ambient space of hyperbolic manifold. Notice the outer shell (ring) with isotropic density along circumference which has the most density, indicating we attained a full tree depth due to the highest density close to the boundary of the Poincaré ball. In contrast, the density (grey regions) in Fig. 4d show the dimensional collapse as encoder ends up producing partially empty ambient space with a reduced Effective Rank of ambient features. The actual feature collapse is caused by the encoder and that is why monitor the ambient space.

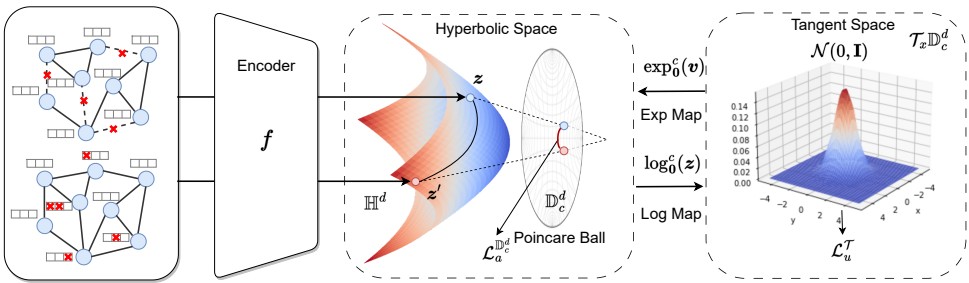

Figure 5: Overall Framework of our Hyperbolic Graph Contrastive Learning (HyperGCL).

### 3.1 APPLYING ALIGNMENT AND OUTER SHELL ISOTROPY FOR HYPERBOLIC LEARNING

**Overall Framework (Figure 5).** We propose Hyperbolic Graph Contrastive Learning (HyperGCL) that leverages the theoretical benefits of the hyperbolic space to impose an isotropic shell level-wise in the ambient space and alignment directly in the hyperbolic space. Appendix A gives our notations.

To generate two augmented graph views, we randomly drop edges and nodes in the graph. The two augmented graph views $(\boldsymbol{X}, \boldsymbol{A})$ and $(\boldsymbol{X}', \boldsymbol{A}')$ are fed into graph encoder $f_\Theta$. $\boldsymbol{Z}$ and $\boldsymbol{Z}'$ are output node embeddings for the two augmented graph views, and $f_\Theta$ is a GCN encoder with two layers:

$$\boldsymbol{Z} = f_\Theta(\boldsymbol{X}, \boldsymbol{A}) = \mathrm{GCN}_2\left(\mathrm{GCN}_1(\boldsymbol{X}, \boldsymbol{A}), \boldsymbol{A}\right) \ \ \text{where} \ \ \mathrm{GCN}_l(\boldsymbol{X}, \boldsymbol{A}) = \sigma\left(\hat{\boldsymbol{D}}^{-\frac{1}{2}}\hat{\boldsymbol{A}}\hat{\boldsymbol{D}}^{-\frac{1}{2}}\boldsymbol{X}\Theta\right). \quad (3)$$

Moreover, $\tilde{\boldsymbol{A}} = \hat{\boldsymbol{D}}^{-1/2}\hat{\boldsymbol{A}}\hat{\boldsymbol{D}}^{-1/2} \in \mathbb{R}^{N \times N}$ is the degree-normalized adjacency matrix, $\hat{\boldsymbol{D}} \in \mathbb{R}^{N \times N}$ is the degree matrix of $\hat{\boldsymbol{A}} = \boldsymbol{A} + \boldsymbol{I}$ where $\boldsymbol{I}$ is the identity matrix, $\boldsymbol{X} \in \mathbb{R}^{N \times d_x}$ contains the initial node features, $\Theta \in \mathbb{R}^{d_x \times d_h}$ contains network parameters, and $\sigma(\cdot)$ is a parametric ReLU (PReLU).

The $N$ pairs of node embeddings, $\{\boldsymbol{z}_i\}_{i=1}^N$, $\{\boldsymbol{z}_i'\}_{i=1}^N$, are firstly projected into the he Poincaré ball $\mathbb{D}_c^d$ as in Appendix B. We set a small margin $\epsilon > 0$ to prevent infinite volume of the hyperbolic manifold.

The goal of Hyperbolic Graph contrastive learning (HyperGCL) is to find the representation to minimize the distance between different augmented embeddings $\boldsymbol{z}_i$ and $\boldsymbol{z}_i'$ of the same sample node $i$ by maximizing hyperbolic alignment $\mathcal{L}_A^{\mathbb{D}_c^d}$ and increase the diversity between distinct node embedding $\boldsymbol{z}_i$ and $\boldsymbol{z}_j$ via improving the outer shell isotropy $\mathcal{L}_A^{\mathbb{D}_c^d}$. We define our new alignment and the outer shell isotropy loss terms and propose to optimize the following objective:

$$\mathcal{L}_{HyperGCL}^{\mathbb{D}_c^d}(\boldsymbol{Z}, \boldsymbol{Z}') = \mathcal{L}_A^{\mathbb{D}_c^d}(\boldsymbol{Z}, \boldsymbol{Z}') + \lambda \cdot \mathcal{L}_U^{\mathbb{D}_c^d}(\boldsymbol{Z}, \boldsymbol{Z}'). \quad (4)$$

In what follows, our focus is on how to design the alignment and especially the outer shell isotropy for the contrastive hyperbolic learning to capture the hierarchical data-invariant information and prevent the dimensional collapse of the ambient space of the hyperbolic manifold.

**Optimizing Alignment in $\mathbb{D}_c^d$.** Hyperbolic spaces are not vector spaces in a traditional sense so Euclidean operations are not applicable. Instead, the formalism of Möbius gyrovector spaces helps generalize standard operations to hyperbolic spaces. We simply define[3] the alignment of $\boldsymbol{z}_i$ and $\boldsymbol{z}_i'$ as:

$$\mathcal{L}_A^{\mathbb{D}_c^d}(\boldsymbol{Z}, \boldsymbol{Z}') = \frac{1}{N}\sum_{i=1}^N D_c\left(\boldsymbol{z}_i, \boldsymbol{z}_i'\right) = \frac{2}{N\sqrt{c}}\sum_{i=1}^N \tanh^{-1}\left(\sqrt{c}\left\|-\boldsymbol{z}_i \oplus \boldsymbol{z}_i'\right\|\right). \quad (5)$$

**Optimizing the Outer Shell Isotropy.** Alleviating the dimensional collapse in the hyperbolic space is not an obvious task. One might naively change the Euclidian distance in Eq. (1) to the hyperbolic distance, *i.e.*, $D_c(\boldsymbol{z}_i, \boldsymbol{z}_i')$ as

$$\mathcal{L}_U^{\mathbb{D}_c^d} = \log \mathop{\mathbb{E}}_{\substack{\text{i.i.d} \\ \boldsymbol{z}, \boldsymbol{z}^- \sim p_Z}}\left[e^{-t D_c(\boldsymbol{z}, \boldsymbol{z}^-)}\right]. \quad (6)$$

However, unlike the Euclidean space where points are constrained on the hypersphere (Eq. (1)), hyperbolic space is non-compact and has infinite volume. Thus, maximizing the pair-wise hyperbolic

---

[3]Kindly note we do not claim the use of Hyperbolic distance as a contribution.

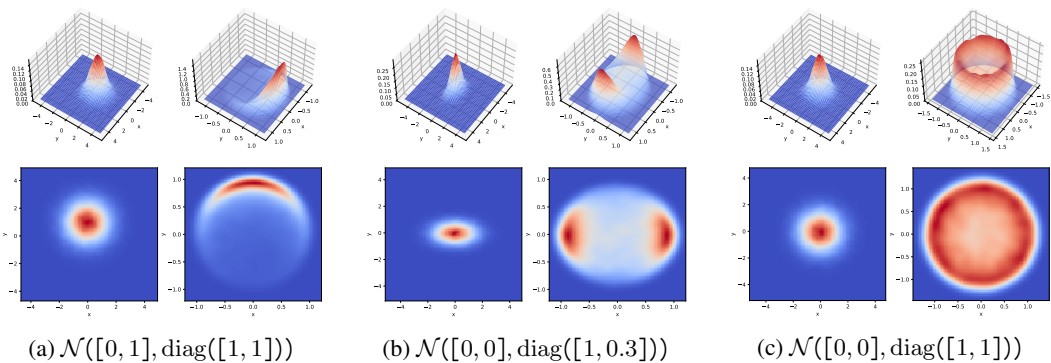

(a) $\mathcal{N}([0, 1], \mathrm{diag}([1, 1]))$    (b) $\mathcal{N}([0, 0], \mathrm{diag}([1, 0.3]))$    (c) $\mathcal{N}([0, 0], \mathrm{diag}([1, 1]))$

Figure 6: Visualization: leaf-level uniformity w.r.t. non-isotropic/non-zero mean Normal distribution.

distance of pairs of points in hyperbolic space via Eq. (6) will push embedding towards the boundary of the Poincare Disk. It will also result in poor "filling" of the ambient space due to the infinite volume at the boundaries, resulting in the height and leaf collapse as in Figure 4d and suboptimal performance (Table 3). Moreover, computing pair-wise distance via Eq. (6) incurs large computation overheads. Due to the exponential growth of volume of hyperbolic manifold, the desired leaf- and height-level uniformity are required. Our investigation reveals that employing an isotropic Normal distribution with zero mean, denoted as $\mathcal{N}(\mathbf{0}, \boldsymbol{I})$, in the tangent plane at $\mathbf{0}$, $\mathcal{T}_{\mathbf{0}}\mathbb{D}_c^n$, yields satisfactory outer shell isotropy in the ambient space of the hyperbolic manifold, as depicted in Figure 6c.

Thus, we propose to impose the feature distribution of learned representations with the ideal Normal distribution in the tangent plane. We gather a set of data vectors from the learned representations in one view, denoted as $\{z_i\}_{i=1}^N \in \mathbb{D}_c^d$, and calculate the mean and covariance matrix as follows:

$$\boldsymbol{\mu} = \frac{1}{N}\sum_{i=1}^N \log_{\mathbf{0}}^c(z_i), \quad \boldsymbol{\Sigma} = \frac{1}{N}\sum_{i=1}^N \left[\log_{\mathbf{0}}^c(z_i) - \boldsymbol{\mu}\right]^T \left[\log_{\mathbf{0}}^c(z_i) - \boldsymbol{\mu}\right], \tag{7}$$

where $\boldsymbol{\mu} \in \mathbb{R}^d$, $\boldsymbol{\Sigma} \in \mathbb{S}_{++}^d$ and $d$ is the dimension of vectors. Let $\boldsymbol{y} = \log_{\mathbf{0}}^c(z) \in \mathbb{R}^d$ denote embedding in the he tangent plane at $\mathbf{0}$. To facilitate the calculation of distribution distance, we apply a Gaussian hypothesis to learned representations where $\boldsymbol{y} \sim \mathcal{N}(\boldsymbol{\mu}, \boldsymbol{\Sigma})$ and

$$q(\boldsymbol{y}) = \frac{1}{(2\pi)^{d/2}\det(\boldsymbol{\Sigma})^{1/2}} \exp\left(-\frac{1}{2}(\boldsymbol{y} - \boldsymbol{\mu})^T \boldsymbol{\Sigma}^{-1}(\boldsymbol{y} - \boldsymbol{\mu})\right). \tag{8}$$

Based on the above assumption, we employ KL divergence, a well-known distance for multivariate Normal distributions, to calculate the distance between two distributions $p$ and $q$:

$$D_{KL}(p, q) = D(\boldsymbol{\Sigma}, \boldsymbol{\mu}) = \frac{1}{2}\left[\mathrm{tr}\left(\boldsymbol{I}^{-1}\boldsymbol{\Sigma}\right) + (\mathbf{0} - \boldsymbol{\mu})^T \boldsymbol{I}^{-1}(\mathbf{0} - \boldsymbol{\mu}) - d + \log\frac{\det \boldsymbol{I}}{\det \boldsymbol{\Sigma}}\right]$$
$$= \mathrm{tr}(\boldsymbol{\Sigma}) - \log\det(\boldsymbol{\Sigma}) - d + \|\boldsymbol{\mu}\|_2. \tag{9}$$

Then changing $\mathcal{L}_U^{\mathbb{D}_c^d}$ in Eq. (4) to the outer shell isotropy term by computing $D(\boldsymbol{\Sigma}, \boldsymbol{\mu}) + D(\boldsymbol{\Sigma}', \boldsymbol{\mu}')$ and removing $-2d$ constant yields:

$$\mathcal{L}_U^{\mathcal{T}} = \mathrm{tr}\left(\boldsymbol{\Sigma} + \boldsymbol{\Sigma}'\right) - \log\det\left(\boldsymbol{\Sigma}\boldsymbol{\Sigma}'\right) - 2d + \|\boldsymbol{\mu}\|^2 + \|\boldsymbol{\mu}'\|^2. \tag{10}$$

## 4 THEORETICAL ANALYSIS

### 4.1 ANALYTICAL MAPPING OF THE NORMAL DISTRIBUTION INTO THE AMBIENT SPACE OF THE HYPERBOLIC MANIFOLD.

**Theorem 2** *The Normal distribution can be mapped into the ambient space of the Hyperbolic manifold by the change of variable. Let the transformation variable be $f(\boldsymbol{v}) = \exp_{\mathbf{0}}^c(\boldsymbol{v}) = \tanh(\sqrt{c}\|\boldsymbol{v}\|)\frac{\boldsymbol{v}}{\sqrt{c}\|\boldsymbol{v}\|_2}$. Then by mapping the Normal distribution from the tangent plane at $\mathbf{0}$ via $f(\boldsymbol{v})$ to the ambient space, we obtain:*

$$p_Z(\mathbf{z}) = 0.5\, p_{\mathcal{N}}(\log_{\mathbf{0}}^c(z); \boldsymbol{\mu}, \boldsymbol{\Sigma}) \cdot \lambda_{\mathbf{z}}^c\, g^{d-1}(\mathbf{z}), \tag{11}$$

where $\lambda_{\mathbf{z}}^c = \frac{2}{1-c\|\mathbf{z}\|_2^2}$ *is the so-called conformal factor and* $g(\mathbf{z}) = \frac{1}{\sqrt{c}\|\mathbf{z}\|_2} \tanh^{-1}(\sqrt{c}\|\mathbf{z}\|_2)$ *and* $p_{\mathcal{N}}(\cdot)$ *is the probability density function (PDF) of the Normal distribution.*

***Proof 1*** *See Appendix C.*

Figure 7 shows the analytical distribution in Eq. (11). Notice its agreement with the simulations in Figure 6. We have also verified that the integral of the distribution within the support region equals 1.

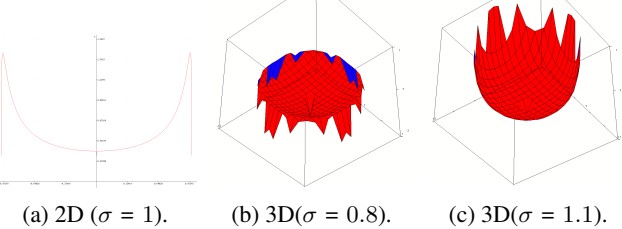

(a) 2D ($\sigma = 1$).  (b) 3D($\sigma = 0.8$).  (c) 3D($\sigma = 1.1$).

Figure 7: The theoretical distribution $p_Z(\mathbf{z})$ of $\mathbf{z}$ in the Hyperbolic space using Eq. (11). The rugged boundary effects in 3D plots are due to poor-quality surface plotting method.

### 4.2 BOUND ON THE EFFECTIVE RANK IN THE TANGENT SPACE

Below we show that minimizing $\mathcal{L}_U^{\mathcal{T}}$ increases the Effective Rank which alleviated the dimensional collapse. Effective Rank in the tangent space measures the effective dimension of embedding output by the encoder $f_\Theta$ in the tangent space. The higher effective rank denotes the lower degree of dimensional collapse. As Effective Rank in the ambient space of the hyperbolic manifold is correlated (just a non-linear mapping) with the Effective Rank in the tangent space, we can directly show how imposing the Normal distribution in the tangent space increase the effective rank.

**Definition 4 (Effective Rank.)** *Let matrix* $\mathbf{X} \in \mathbb{R}^{m \times n}$ *with* $\mathbf{X} = \mathbf{U}\mathbf{\Sigma}\mathbf{V}^T$ *as its singular value decomposition, where* $\mathbf{\Sigma}$ *is a diagonal matrix with singular values* $\sigma_1 \geq \cdots \geq \sigma_Q \geq 0$ *with* $Q = \min(m, n)$. *The distribution of singular values is defined as the normalized form* $p_i = \sigma_i / \sum_{k=1}^Q |\sigma_k|$. *The effective rank of the matrix* $\mathbf{X}$, *is defined as* $\mathrm{Erank}(\mathbf{X}) = \exp\left(H\left(p_1, p_2, \cdots, p_Q\right)\right)$, *where* $H\left(p_1, p_2, \cdots, p_Q\right)$ *is the Shannon entropy given by* $H\left(p_1, p_2, \cdots, p_Q\right) = -\sum_{k=1}^Q p_k \log p_k$.

**Theorem 3** *The Effective Rank of* $\mathbf{\Sigma}$ *is lower-bounded by* $-D(\mathbf{\Sigma}, \boldsymbol{\mu})$ *as:*

$$-D(\mathbf{\Sigma}, \boldsymbol{\mu}) \leq \log\left[\mathrm{Erank}(\mathbf{\Sigma})\right] + const \tag{12}$$

*Let* $\{\lambda\}_{i=1}^d$ *be the eigenvalues of* $\mathbf{\Sigma}$, *the equality is hold when* $\lambda_i = \lambda_j$ *for all* $i, j$.

**Proof 2** *Proof of Lemma 3 is in Appendix D.*

Theorem 3 implies that the sum of the effective rank of $\mathbf{\Sigma}$ and $\mathbf{\Sigma}'$, *i.e.*, $\mathrm{Erank}(\mathbf{\Sigma}) + \mathrm{Erank}(\mathbf{\Sigma}')$ is lower-bounded by the proposed outer isotropy term $-\mathcal{L}_U^{\mathcal{T}} = -(D(\mathbf{\Sigma}, \boldsymbol{\mu}) + D(\mathbf{\Sigma}', \boldsymbol{\mu}'))$. Minimizing $\mathcal{L}_U^{\mathcal{T}}$ achieves a higher effective rank which implies a lower degree of dimensional collapse.

## 5 EXPERIMENTS

### 5.1 RESULTS ON GRAPH REPRESENTATION LEARNING

**Datasets.** We use the citation networks **Cora**, **CiteSeer** and **Pubmed** whose nodes represent scientific papers (Kipf & Welling, 2016). We also use the **Disease** dataset whose node labels tell whether the node was infected or not. **Airport** is a dataset with nodes and edges representing airports and routes from OpenFlights.org (Zhang & Chen, 2018). In the node classification (NC), we use 70/15/15% splits for the Airport dataset, 30/10/60% splits for Disease, and we use standard splits (Kipf & Welling, 2016) with 20 train examples per class for Cora, CiteSeer and PubMed. As in previous works (Veličković et al., 2017), we evaluate node classification by measuring accuracy.

**Setting and Baselines.** Below we investigate both Euclidean embeddings and hyperbolic embeddings. Our model is compared in (i) self-supervised and (ii) supervised (model trained with labeled data) settings. In the self-supervised setting, we follow the linear evaluation from Veličković et al. (2017) and Zhu et al. (2020), where models are first trained in an unsupervised manner. Subsequently, the node representations are fed and evaluated with a logistic regression classifier on labeled data. Note that HyperGCL operates in a self-supervised setting. For Euclidean graph embeddings, we compare

Table 1: Comparison with various node classification models. ♥ marks the methods that are trained in the supervised manner and ♦ marks the methods that are trained in the self-supervised manner.

| Space | Method | Disease | Airport | PubMed | CiteSeer | Cora |
|---|---|---|---|---|---|---|
| Euclidean | ♥ GCN | 69.70±0.40 | 81.40±0.60 | 81.10±0.20 | 70.41±0.52 | 81.30±0.30 |
| | ♥ GAT | 70.40±0.40 | 81.50±0.30 | 82.00±0.30 | 70.14±0.38 | 83.00±0.70 |
| | ♥ SAGE | 69.10±0.60 | 82.10±0.50 | 80.40±2.20 | 69.91±1.38 | 77.90±2.40 |
| | ♦ GRACE | 69.61±0.49 | 82.79±0.40 | 83.51±0.37 | 71.42±0.64 | 81.13±0.44 |
| | ♦ COSTA | 67.12±0.39 | 81.19±0.40 | 84.31±0.37 | 70.77±0.24 | 82.14±0.62 |
| Hyperbolic | ♥ HNN | 75.18±0.25 | 80.59±0.46 | 76.88±0.43 | 59.50±1.28 | 54.76±0.61 |
| | ♥ HGNN | 81.27±1.53 | 84.71±0.98 | 80.13±0.82 | 69.99±1.00 | 78.26±1.19 |
| | ♥ HGCN | 88.16±0.76 | 89.26±1.27 | 82.53±0.63 | 70.34±0.59 | 78.03±0.98 |
| | ♥ HGAT | 90.30±0.62 | 89.62±1.23 | 81.42±0.36 | 70.64±0.30 | 78.32±1.39 |
| | ♦ HGCL | 93.42±0.82 | 92.35±1.51 | 83.14±0.58 | 72.11±0.64 | 82.37±0.47 |
| | ♦ HyperGCL | **94.50**±0.43 | **93.55**±1.11 | **85.14**±0.38 | **73.43**±0.35 | **84.47**±0.46 |

Table 2: Comparison with various competing models. ♠ denotes methods that are designed in the Euclidean space and ♣ denotes methods that are designed in the hyperbolic space.

| Datasets | | ♠NGCF | ♠LGCN | ♣HAE | ♣HAVE | ♣HGCF | ♣HRCF | ♣HyperGCL |
|---|---|---|---|---|---|---|---|---|
| **Amazon-CD** | R@10 | 0.0758 | 0.0929 | 0.0666 | 0.0781 | 0.0962 | 0.1003 | **0.1069** |
| | R@20 | 0.1150 | 0.1404 | 0.0963 | 0.1147 | 0.1455 | 0.1503 | **0.1573** |
| | N@10 | 0.0591 | 0.0726 | 0.0565 | 0.0629 | 0.0751 | 0.0785 | **0.0825** |
| | N@20 | 0.0718 | 0.0881 | 0.0657 | 0.0749 | 0.0909 | 0.0947 | **0.1043** |
| **Amazon-Book** | R@10 | 0.0658 | 0.0799 | 0.0634 | 0.0774 | 0.0867 | 0.0900 | **0.0973** |
| | R@20 | 0.1050 | 0.1248 | 0.0912 | 0.1125 | 0.1318 | 0.1364 | **0.1489** |
| | N@10 | 0.0655 | 0.0780 | 0.0709 | 0.0778 | 0.0869 | 0.0902 | **0.0982** |
| | N@20 | 0.0791 | 0.0938 | 0.0789 | 0.0901 | 0.1022 | 0.1060 | **0.1060** |
| **Yelp2020** | R@10 | 0.0458 | 0.0522 | 0.0360 | 0.0421 | 0.0527 | 0.0537 | **0.0587** |
| | R@20 | 0.0764 | 0.0866 | 0.0588 | 0.0691 | 0.0884 | 0.0898 | **0.0950** |
| | N@10 | 0.0405 | 0.0461 | 0.0331 | 0.0371 | 0.0458 | 0.0468 | **0.0508** |
| | N@20 | 0.0513 | 0.0582 | 0.0409 | 0.0465 | 0.0585 | 0.0594 | **0.0639** |

HyperGCL against several models. In the supervised setting, we consider GCN (Kipf & Welling, 2016), GAT (Veličković et al., 2017), SGAE (Hamilton et al., 2017). In the self-supervised setting, we compare HyperGCL with GRACE (Zhu et al., 2020) and COSTA (Zhang et al., 2022). For hyperbolic graph embeddings, we compare HyperGCL with the supervised hyperbolic models HGCN (Chami et al., 2019), HGAT (Zhang et al., 2021), HGNN (Liu et al., 2019c). We also compare HyperGCL with the self-supervised hyperbolic model HGCL (Liu et al., 2022).

**Results.** Table 1 compares HyperGCL to baseline methods. HyperGCL achieves the highest performance in both the supervised GNN and self-supervised GCL settings. Notably, the performance gains of HyperGCL over Euclidean GNN models show that graph contrastive learning can benefit from the hyperbolic geometry. Furthermore, HyperGCL outperforms HGCL, indicating the effectiveness of our proposed hyperbolic alignment and outer shell isotropy losses.

## 5.2 Result of Representation Learning in Collaborative Filtering

In large-scale recommender systems, user-item relationships often exhibit scale-free or exponential expansion characteristics, making them particularly suitable for hyperbolic embedding and adding the outer shell isotropy loss into the HRCF (Yang et al., 2022). Appendix E provides detailed setting.

**Results.** Table 2 shows that HyperGCL outperforms the baseline models on all datasets across both Recall@20 and NDCG@20. We notice that hyperbolic models equipped with the ranking loss (*i.e.*, HGCF and HRCF) show a significant advantage over their Euclidean counterparts (*i.e.*, LGCN), demonstrating the superiority of hyperbolic geometry for modeling user-item networks. Using $\mathcal{L}_U^{\mathbb{D}_c^d}$ to adjust the embedding space also improves the performance by alleviating the dimensional collapse.

## 5.3 Analysis

**Ablation Study.** Below we analyze various components of HyperGCL in Table 3, which shows that applying $\mathcal{L}_U^{\mathbb{R}^d} / \mathcal{L}_A^{\mathbb{R}^d}$ in the tangent space or applying $\mathcal{L}_U^{\mathbb{D}_c^d}/\mathcal{L}_A^{\mathbb{D}_c^d}$ in the hyperbolic space does not achieve higher effective rank, and yields low performance compared to our method. There is nothing to prevent the network from learning a constant embedding for all nodes if only the alignment loss (the third row) is used. While applying augmentation alleviates it from the full collapse, it achieves a lower effective rank. Our HyperGCL (last row) achieves a higher effective rank and the best results.

**The Impact of Gaussian Centering and Non-isotropy (tangent space).** Below we explore the impact of these above two factors on the outer shell isotropy loss. We scale vector **1** by some constant

Table 3: Ablations on HyperGCL. Erank of embedding is measured in the ambient space (and (Erank) in the tangent space) of the encoder output. Eranks in the ambient and tangent spaces correlate well.

| Manifold | Align | Uni | PubMed | | CiteSeer | | Cora | | Disease | | Airport | |
|---|---|---|---|---|---|---|---|---|---|---|---|---|
| | | | Acc. | Erank | Acc. | Erank | Acc. | Erank | Acc. | Erank | Acc. | Erank |
| Euclidean | $\mathcal{L}_A^{\mathbb{R}^d}$ | $\mathcal{L}_U^{\mathbb{R}^d}$ | 83.14±0.18 | 5.22(5.20) | 71.43±0.52 | 23.01(23.03) | 82.37±0.27 | 5.50(5.47) | 73.40±0.24 | 2.19 (2.15) | 81.30±0.21 | 2.34(2.32) |
| Tangent | $\mathcal{L}_A^{\mathbb{R}^d}$ | $\mathcal{L}_U^{\mathbb{R}^d}$ | 82.34±0.35 | 4.79(4.78) | 71.42±0.67 | 22.87(23.02) | 81.34±0.33 | 4.94(4.93) | 69.42±0.45 | 2.08(2.05) | 79.53±0.41 | 2.02(2.01) |
| Hyperbolic | $\mathcal{L}_A^{\mathbb{D}_C^d}$ | × | 71.02±0.13 | 1.22(1.19) | 63.84±0.43 | 5.93(5.92) | 72.27±0.12 | 1.23(1.22) | 42.40±0.64 | 1.02(1.01) | 52.31±0.31 | 1.06(1.05) |
| Hyperbolic | $\mathcal{L}_A^{\mathbb{D}_C^d}$ | $\mathcal{L}_U^{\mathbb{D}_C^d}$ | 81.51±0.37 | 4.43(4.42) | 70.23±0.41 | 21.01(21.01) | 81.13±0.44 | 4.50(4.47) | 90.30±0.82 | 3.59(3.58) | 91.70±0.71 | 4.15(4.14) |
| Hyperbolic | $\mathcal{L}_A^{\mathbb{D}_C^d}$ | $\mathcal{L}_U^{\mathcal{T}}$ | 85.14±0.38 | 6.89(6.88) | 73.43±0.66 | 24.75(24.72) | 84.47±0.16 | 7.76(7.75) | 94.50±0.43 | 4.79(4.78) | 93.55±1.11 | 5.25(5.23) |

factor, as shown in Table 4. To check if non-isotropy has a negative impact on results, we randomly sample a fraction of $p = 0.3, 0.5, 0.7$ from the diagonal elements of $\mathbf{I}$ and set them to be 0.01 to simulate the non-isotropic Normal distribution (covariance matrix denoted by $\mathbf{I}_p$). When moving from the zero-mean or violating isotropy of Gaussian, the mode collapse in the hyperbolic space occurs, as in Figure 6a. Table 4 confirms the detrimental impact of non-isotropy and non-zero centering on experimental results.

Table 4: Test performance w.r.t. the isotropic Normal distribution with different mean centers.

| Gaussian | PubMed | | CiteSeer | | Cora | |
|---|---|---|---|---|---|---|
| | Acc. | Erank | Acc. | Erank | Acc. | Erank |
| $\mathcal{N}(0.0 \cdot \mathbf{1}, \mathbf{I})$ | 85.14±0.38 | 6.89 | 73.43±0.64 | 24.75 | 84.47±0.16 | 7.76 |
| $\mathcal{N}(0.5 \cdot \mathbf{1}, \mathbf{I})$ | 80.13±0.24 | 4.27 | 69.43±0.22 | 18.34 | 80.27±0.11 | 4.31 |
| $\mathcal{N}(1.0 \cdot \mathbf{1}, \mathbf{I})$ | 80.79±0.42 | 4.20 | 70.25±0.15 | 20.65 | 79.47±0.35 | 4.45 |
| $\mathcal{N}(\mathbf{0}, \mathbf{I}_{0.3})$ | 84.24±0.18 | 5.59 | 71.82±0.47 | 20.41 | 82.32±0.13 | 4.98 |
| $\mathcal{N}(\mathbf{0}, \mathbf{I}_{0.5})$ | 82.34±0.23 | 5.02 | 70.13±0.34 | 16.32 | 80.17±0.15 | 4.20 |
| $\mathcal{N}(\mathbf{0}, \mathbf{I}_{0.7})$ | 78.24±0.64 | 4.15 | 67.43±0.35 | 9.69 | 75.25±0.14 | 2.85 |

**Discussion on Erank**. Table 3 reveals that Effective Ranks in the ambient space and the tangent space are highly correlated (as expected as these spaces are connected via non-linear mapping). Theorem 3 shows that imposing the isotropic Gaussian in the tangent space on features improves its effective rank. Ergo, we improve the effective rank in the ambient space by promoting the outer shell isotropy in the ambient space which means preventing the leaf and height collapse.

## 6 RELATED WORKS

**Hyperbolic Graph Neural Networks.** Recent works (Liu et al., 2019b; Chami et al., 2019; Zhang et al., 2021; He et al., 2020; Liu et al., 2022) extend GNNs to the hyperbolic space. HGNN (Liu et al., 2019b), HGCN (Chami et al., 2019), and HGAT (Zhang et al., 2021) use graph convolutions in the tangent space. LGCN (He et al., 2020) proposes graph convolutions on the hyperbolic manifold. HGNN (Liu et al., 2019b) (graph classification) provides an extension to dynamic graph embeddings. HGAT (Zhang et al., 2021) (node classification and clustering) introduces a hyperbolic attention-based graph convolution. HGCN (Chami et al., 2019) develops a learnable curvature model. LGCN (He et al., 2020) aggregates the neighborhood information via centroid of Lorentzian distance. HGCL (Liu et al., 2022) utilizes contrastive learning to improve the hyperbolic graph learning.

**Graph Contrastive Learning.** Inspired by CL methods in vision and NLP (Chen et al., 2020a; Gao et al., 2021), CL has also been adapted to the graph domain. By adapting DeepInfoMax (Bachman et al., 2019) to graph representation learning, DGI (Velickovic et al., 2019) learns embedding by maximizing the mutual information to discriminate between nodes of original and corrupted graphs. REFINE (Zhu & Koniusz, 2021) uses a simple negative sampling term inspired by skip-gram models. Fisher-Bures Adversarial GCN (Sun et al., 2019) uses adversarial perturbations of graph Laplacian. Inspired by SimCLR (Chen et al., 2020a), GRACE (Zhu et al., 2020) correlates graph views for node-level task. while GraphCL method (Hafidi et al., 2020) learns embeddings for graph-level tasks.

## 7 CONCLUSIONS

We believe ours is the first to investigate the notion of dimensional collapse for hyperbolic graph embedding. We have discussed that poor quality hyperbolic embedding space results in the leaf- and height-level collapse of tree-equivalent cases. To that end, we have shown that imposing a zero-centered isotropic Gaussian distribution in the tangent plane at $\boldsymbol{x} = \mathbf{0}$, $\mathcal{T}_{\boldsymbol{x}}\mathbb{D}_c^n$, results in the best levels of leaf- and height-level uniformity in the hyperbolic spaces. Such a notion uniformity translates into the outer shell isotropy in the ambient space of the hyperbolic manifold. Moreover, by employing KL-divergence, we have shown how such a zero-centered isotropic Gaussian distribution can be mapped to the Poincaré disk via the exponential map.

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

# ALIGNMENT AND OUTER SHELL ISOTROPY FOR HYPERBOLIC GRAPH CONTRASTIVE LEARNING −APPENDICES−

**Anonymous authors**

## A NOTATIONS

**Notations.** In this paper, a graph with node features is denoted as $G = (\mathbf{X}, \mathbf{A})$ and $\mathbf{X} \in \mathbb{R}^{N \times d_x}$ is the feature matrix (*i.e.*, the $i$-th row of $\mathbf{X}$ is the feature vector $\boldsymbol{x}_i$ of node $v_i$) and $\mathbf{A} \in \{0, 1\}^{n \times n}$ denotes the adjacency matrix of $G$, *i.e.*, the $(i, j)$-th entry in $\mathbf{A}$ is 1 if there is an edge between nodes $i$ and $j$. The degree of node $i$, denoted as $d_i$, is the number of edges incident with $i$. The degree matrix $\mathbf{D}$ is a diagonal matrix and its $i$-th diagonal entry is $d_i$. For a $d$-dimensional vector $\boldsymbol{x} \in \mathbb{R}^d$, $\|\boldsymbol{x}\|_2$ is the Euclidean norm of $\boldsymbol{x}$. We use $x_i$ to denote the $i$ th entry of $\boldsymbol{x}$, and $x_{ij}$ for the $(i, j)$-th entry of $\mathbf{X}$. $\operatorname{diag}(\boldsymbol{x}) \in \mathbb{R}^{d \times d}$ is a diagonal matrix such that the $i$-th diagonal entry is $x_i$. We use $\boldsymbol{x}_i$ denote the row vector of $\mathbf{X}$. The trace of a square matrix $\mathbf{X}$ is denoted by $\operatorname{tr}(\mathbf{X})$, which is the sum along the diagonal of $\mathbf{X}$.

## B PROJECTION INTO THE POINCARÉ BALL

Assume the output space of the graph neural network $f_\Theta(\cdot)$ is in the Poincaré ball $\mathbb{D}_c^d$, we project the all the node embedding to the $\mathbb{D}_c^d$ as

$$\boldsymbol{z} := \begin{cases} \boldsymbol{z} & \text{if } \|\boldsymbol{z}\| \le \frac{1}{c} \\ (1 - \epsilon)\frac{\boldsymbol{z}}{c\|\boldsymbol{z}\|} & \text{else} \end{cases} \tag{13}$$

## C PROOF OF THEOREM 2

It directly follow from the transformation of random variables. Specifically, $p_Z(\mathbf{z}) = p_{\mathcal{N}}(f^{-1}(\mathbf{z})) \cdot \det\left(\mathbf{J}(f^{-1}(\mathbf{z}))\right)$. Notice that for $f(\boldsymbol{v}) = \exp_{\mathbf{0}}^c(\boldsymbol{v})$ the inverse is logarithmic map $f^{-1}(\mathbf{z}) = \log_{\mathbf{0}}^c(\mathbf{z}) = \frac{1}{\sqrt{c}\|\mathbf{z}\|_2} \tanh^{-1}(\sqrt{c}\|\mathbf{z}\|_2)\frac{\mathbf{z}}{\sqrt{c}\|\mathbf{z}\|_2}$. The main difficulty lies with computing the Jacobian $\mathbf{J}(f^{-1}(\mathbf{z}))$ and its determinant $\det\left(\mathbf{J}(f^{-1}(\mathbf{z}))\right)$, which (after crunching some maths) turns out to enjoy a simple analytical form $0.5 \lambda_{\mathbf{z}}^c g^{d-1}(\mathbf{z})$.

## D PROOF OF LEMMA 3

$$D(\boldsymbol{\Sigma}, \boldsymbol{\mu}) = \operatorname{tr}(\boldsymbol{\Sigma}) - \log \det(\boldsymbol{\Sigma}) - d = \sum_{i=1}^d (\lambda_i - \log \lambda_i - 1). \tag{14}$$

We usually centralize the embedding $\{log_{\mathbf{0}}^c(\boldsymbol{z}_i)\}$, therefore we ignore the $\boldsymbol{u}$ for brevity in Eq. (14). We know that $x - \log x \ge 1$ with equality at $x = 1$. and $x - \log x \ge \log x + 1 - \log 4$ with equality at $x = 2$. Given $\lambda_1 \ge \lambda_2 \cdots \ge \lambda_d > 0$, we have:

$$D(\boldsymbol{\Sigma}, \boldsymbol{\mu}) \ge (\log \lambda_1 + 1 - \log 4) - (\log \lambda_d + 1)$$
$$D(\boldsymbol{\Sigma}, \boldsymbol{\mu}) \ge (\log \lambda_1 - \log \lambda_d) + const$$
$$D(\boldsymbol{\Sigma}, \boldsymbol{\mu}) \ge \left(\log \frac{\lambda_1}{\lambda_d}\right) + const = \log \frac{\lambda_1}{\lambda_2} + \log \frac{\lambda_2}{\lambda_3} \cdots \log \frac{\lambda_{d-1}}{\lambda_d} + \log \frac{\lambda_d}{\lambda_d} + const. \tag{15}$$

Let $q_i = \frac{\lambda_i}{\sum_i^d \lambda_i}$ and $0 < q_i \leq 1$, then:

$$
\begin{aligned}
D(\boldsymbol{\Sigma}, \boldsymbol{\mu}) &= \log \frac{q_1}{q_2} + \log \frac{q_2}{q_3} + \log \frac{q_{d-1}}{q_d} + \log \frac{q_d}{q_d} + const \\
&\geq \log q_1 + \log q_2 \cdots \log q_{d-1} + \log q_d + const \\
&\geq \sum_{q=1}^{d} q_i \log q_i + const \\
-D(\boldsymbol{\Sigma}, \boldsymbol{\mu}) &\leq - \sum_{q=1}^{d} q_i \log q_i + const \\
\exp(-D(\boldsymbol{\Sigma}, \boldsymbol{\mu})) &\leq \exp(- \sum_{q=1}^{n-1} q_i \log q_i) + const \\
D(\boldsymbol{\Sigma}, \boldsymbol{\mu}) &\geq - \log \mathrm{Erank}(\boldsymbol{\Sigma}) + const.
\end{aligned}
\tag{16}
$$

Thus, our loss yields an upper bound on the Effective Rank.

## E    SETTING OF THE COLLABORATIVE FILTERING

**Datasets.** We use three publicly available datasets Amazon-Book, Amazon-CD, and Yelp2020, which are also employed in the HRCF. The statistics are summarized in Table 5 in the appendix.

**Baselines.** Compared methods. To verify the effectiveness of our proposed method, the compared methods include both well-known or leading hyperbolic models and Euclidean baselines. For hyperbolic models, the HGCF (Sun et al., 2021), HVAE and HAE (Liang et al., 2018) and are compared. HAE (HVAE) combines a (variational) autoencoder with hyperbolic geometry. Besides, we include strong Euclidean baselines, *i.e.*, LGCN (He et al., 2020) and NGCF (Wang et al., 2019).

**Setting.** To show that hyperbolic uniformity is crucial for learning the hierarchical representation, we combine the proposed uniformity metric with the existing SOTA method (*i.e.*, HRCF (Yang et al., 2022)) by adding $\mathcal{L}_U^{\mathbb{D}_c^d}$ as an auxiliary loss. We test the model using the relevancy-based metric Recall@20 and the ranking-aware metric NDCG@20. In order to maintain a fair comparison and reduce the workload of our experiments, we closely adhere to the settings of HRCF (Yang et al., 2022). Specifically, we set the embedding size to 50 and fix the total training epochs at 500. The range of $\lambda$ values in the loss function is {10, 15, 20, 25, 30}, while the aggregation order is searched in range from 2 to 10. When it comes to the margin, we explore values within the range of {0.1, 0.15, 0.2}. To train the network parameters, we employ Riemannian SGD (Bonnabel, 2013) with weight decay, using values from the range 1e-4, 5e-4, 1e-3, along with learning rates of {0.001, 0.0015, 0.002}. RSGD is a technique that emulates stochastic gradient descent optimization while accounting for the geometry of the hyperbolic manifold (Bonnabel, 2013). For the baseline settings of HAE, HAVE and HGCF, we refer the reader to (Sun et al., 2021).

Table 5: Statistics of the experimental datasets.

| Dataset | #User | #Item | #Interactions | Density |
|---|---|---|---|---|
| **Amazon-CD** | 22,947 | 18,395 | 422,301 | 0.00100 |
| **Amazon-Book** | 52,406 | 41,264 | 1,861,118 | 0.00086 |
| **Yelp2020** | 71,135 | 45,063 | 1,940,014 | 0.00047 |

## F    IMPACT OF CURVATURE $c$.

Since the curvature parameter $c$ controls the depth of hierarchy (height of the tree embedding), we analyze its effect on results. The notion of height-level uniformity is related to the value of $c$: the larger $c$ is, the more concentration of the distribution towards the tree root. Figure 8 shows results w.r.t. varying $c$. The result shows HyperGCL achieves the best result for different $c$ meaning the the

height-level uniformity is data dependent and related to sparsity of the datasets (sparsity is indicated in caption brackets of Figure 8), *e.g.*, graphs with relatively larger density require smaller $c$.

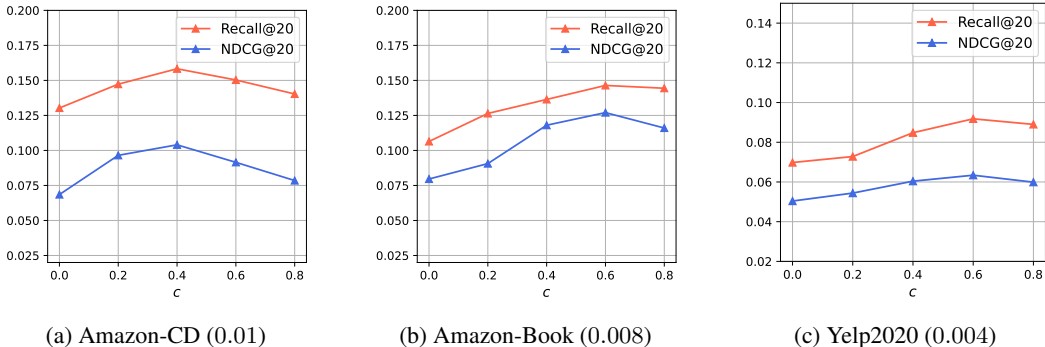

(a) Amazon-CD (0.01)                (b) Amazon-Book (0.008)                (c) Yelp2020 (0.004)

Figure 8: Performance w.r.t. the value of curvature $c$. In caption brackets, we indicate the dataset sparsity.

## G   BROADER IMPACT AND LIMITATIONS

Our method enjoys impact and limitations similar to those in graph contrastive learning. Typical GCL models cannot guarantee they can utilize the feature space efficiently due to the mode collapse phenomenon. As we utilize the feature space more efficiently due to the Hyperbolic geometry and the penalty preventing collapse, our model works better, delivering better prediction on graphs for the similar computational cost. Our idea can be universally applied to other scenarios where the mode collapse is an issue. Of course, in this work we do not study fairness or biases but we believe that poorer utilization of the feature space in other methods can exacerbate such issues.

