# OpenReview forum: "Alignment and Outer Shell Isotropy for Hyperbolic Graph Contrastive Learning"
_ICLR.cc/2024/Conference — Submitted to ICLR 2024_

### Official Review · Reviewer_WCaW · 2023-10-22

**Soundness:** 1 poor
**Presentation:** 1 poor
**Contribution:** 2 fair
**Rating:** 3
**Confidence:** 4

**Summary:**

The authors proposed a contrastive learning method using hyperbolic space with the alignment between representations and the standard normal distribution in the tangent space at the origin of the hyperbolic space. The authors discussed the effectiveness of the proposed method using the Effective rank of the representations in a tangent space.

**Strengths:**

1. The proposed model is simple, intuitive, and novel in the contrastive learning area, as far as I know.
1. The proposed model has shown good results in numerical experiments.

**Weaknesses:**

While main ideas of the paper are interesting to me, the paper needs significant improvement mainly due to its incompleteness and mathematical soundness issues.
1. Contrastive learning, the main theme of the paper, is not defined in the paper. Readers cannot understand what kind of data we are handling and suddenly see the pair of (X, A) and (X', A') in Section 3.1. without knowing what kind of real data these symbols correspond to in real applications.
1. The explanation of Theorem 1 is wrong. Theorem 1 roughly says that distance among points in hyperbolic space can be approximated by a tree, but does NOT say its converse.  However, your explanation is "any tree can be embedded into a
Poincaré disk with low distortion." This is the converse of Theorem 1.
1. Many important definitions are missed. What are $p_\\mathrm{pos}$ and $p_\\mathrm{data}$ in Equation (1)? Also, $p$ and $D$ are not defined in Equation (9).
1. Theorem 3 says nothing since the left-hand side, the negative KL distance, is always non-positive and the right-hand side, Erank, which is the entropy of the normalized singular values, is always non-negative. Also, these are not strongly related. If we consider $\\mathcal{N}(\\mathbf{0},c\\mathbf{I})$, we can see that the entropy of the normalized singular values is constant but the KL divergence varies as we change $c$. For the above reasons, the discussion of the results in Table 3 using Erank, which is the entropy of the normalized singular values of the covariance matrix in the tangent space, does not support the justification of the proposed method. First of all, the authors could have shown the value $\\mathcal{L}_{U}^\\mathcal{T}$ in Table 3, instead of Erank. Hence, the whole structure of the paper needs significant improvement. This technical issue regarding Theorem 3 is the strongest concern about this paper.
1. The motivation of introducing the term $\\mathcal{L}_{U}^\\mathcal{T}$ is to force the uniformity. However, since the normal distribution decays exponentially in the tangent space and the surface area of a hyper-ball in hyperbolic space exponentially grows with respect to its radius $r$, the density of the normal distribution decays double exponentially. So the concentration effect is much more dominant than the uniformity effect. The authors have not discussed it.
1. In Section 3.1., it says "The N pairs of node embeddings,... as in Appendix B." However, the reason is not described.
1. Figure 4 does not make sense since the author states that 4b is the best while 4b ignores the topology of the original tree. If we could ignore the topology of the original graph, we should randomly map them to the lattice in Euclidean space. Also, if we consider the hyperbolic distance, 4d seems more uniform than 4b.
1. The proposed method does not scale well with respect to the dimensionality, since it involves the determinant calculation. It might not be critical in real applications where the dimensionality is not large.

**Questions:**

1. Could you define contrastive learning formally so it is consistent with your proposal method's notation? Also, based on the explanation, could you explain which data in real applications can the contrastive learning apply to?
1. What are $p_\\mathrm{pos}$ and $p_\\mathrm{data}$ in Equation (1)?
1. What does $\\mathcal{T}$ mean in $\\mathcal{L}_{U}^\\mathcal{T}$?

---

> ### Author Response · Authors · 2023-11-22
> **Rebuttal (Reviewer WCaW)**
>
> # 1. Theorem 3 says nothing since the left-hand side, the negative KL distance, is always non-positive and the right-hand side, Erank, is always non-negative.
>
> We politely disagree. **Reviewer overlooks a critical aspect of Theorem 3 by failing to acknowledge the negative constant involved.** According to the proof outlined in **the appendix, the theorem stipulates a negative constant of $\log 4 - 2$**. It is essential to note that Theorem 3 explicitly states that $D(\mathbf{\Sigma, \mu})$ serves as an upper bound for $-\log[Erank(\Sigma)]$. **Thus, considering the negative constant is pivotal to comprehending the theorem's implications accurately.**
>
> # 2. Proposed loss and Erank are not strongly related.
>
> This is not correct. The reviewer selectively highlights the case where Erank has attained its maximum, i.e, when the singular value of $\sigma$ follows a uniform distribution ($\mathcal{N}(0,c\mathbf{I})$). At this point, minimizing the $D(\mathbf{\Sigma, \mu})$ can only result in the reduction of the scaling parameter $c$. Note that once Erank reaches its peak, the model has essentially achieved our objective of rendering the embedding space isotropic. In practice, the model is a trade-off between isotropy and other loss terms so the above example is cherry-picked.
>
> Note that minimizing $D(\mathbf{\Sigma, \mu})$ significantly enhances $Erank$ when the singular values of $\Sigma$ are not uniformly distributed. From an optimization standpoint, we establish that:
> $$
> -logdet(\Sigma) = -\sum_{i=1}^{N} log \sigma_i
> $$
> considering $\text{Tr}(\Sigma) =  \sum_{i}^{N}\sigma_i \leq C$ (while minimizing $\text{Tr}(\Sigma)$), where the function $-\log(\sigma)$ is convex, utilizing Jensen's inequality allows us to conclude that the optimization problem $\min \sum_{i=1}^{N} \log \sigma_i$ attains its optimum when all $\sigma$ values are equal. Solving Eq. 9 through optimization results in a solution with uniform singular values, thereby maximizing the Erank.
>
> # 3. Showing $\mathcal{L}\_{U}^\mathcal{T}$ instead of Erank.
>
> Showing $\mathcal{L}\_{U}^\mathcal{T}$ is not our objective, as we minizie $\mathcal{L}_{U}^\mathcal{T}$ in our loss, it of course will decrease. **The degree of dimensional collapse is measured by Erank.**
>
> # 4. Concentration effect is much more dominant than the uniformity effect. The authors have not discussed it.
>
> What does "the concentration effect is much more dominant than the uniformity effect" mean? Is it a standard terminology? All we claim is the resulting feature distribution becomes as in Fig 6c and 7a. It is clear from 7a and 7b how the concentration of features looks like. We do not see any non-uniformity effect there.
>
> # 5. In Section 3.1., it says "The N pairs of node embeddings,... as in Appendix B." However, the reason is not described.
>
> As our encoder generates embeddings in the hyperbolic space, it is essential to project (constrain) these embeddings onto the Poincaré ball as in many other papers:
> > Kernel Methods in Hyperbolic Spaces, IJCV 2023
>
> # 6. Figure 4 does not make sense: author states that 4b is the best while 4b ignores the topology of the original tree. If we could ignore the topology of the original graph, we should randomly map them to the lattice in Euclidean space.
>
> The analysis does not assume prior knowledge of the original tree structure: why should it? **Fig 4.  represent potential embedding outcomes generated by the algorithm not some one and only solution** with topology A or B. Our objective is to evaluate/compare these figures to discern which one shows a lesser degree of dimensional collapse.
>
> Topology of input graph has little to do with its embeddings. One may embed an images into hyperbolic space which will enjoy the tree structure. too. Does rev. know underlying image space topology to connect both? We do not.
>
> # 7. The proposed method does not scale well w.r.t. the dimensionality, as it involves determinant calculation.
>
> We politely disagree.  Determinant by Cholesky factorization is fast. Overhead is a tiny fraction even with large d. The following table shows time w.r.t. d:
>
> |d|time(ms)|
> |-|-|
> |5| 0.03|
> |10|0.04|
> |20|0.06|
> |40|0.12|
> |80|0.39|
> |200|2.22|
> |500|22.1|
> |1000|119.7|
>
> Determinant is  widely used by many  papers with large $d$ and thousands/millions of samples. It is computationally reasonable choice (**and faster approximations of it exist**):
> > Learning Log-Determinant Divergences for Positive Definite Matrices (TPAMI 2021)
>
> # 8. p and D are not defined in Eq. (9).
> There is no D in Eq. 9. There is d defined at the top of section 2.2. p is also explained in the line above Eq. 9.
>
> # 9. What are $p_{pos}$ and $p_{data} $ in Eq. (1).
>
> These symbols follow standard terminology of Wang & Isola (2020) cited just above Eq. 1. We will explain them accordingly.
>
> # 10. The explanation of Th. 1 is wrong.
>
> We explained how Th. 1 matters to our work, not what original author's intentions were. We are happy to rephrase. A one minute job.

---

> ### Comment · Reviewer_WCaW · 2023-11-22
>
> Thank you for your detailed comments.
>
> - **1.** *Reviewer overlooks a critical aspect of Theorem 3 by failing to acknowledge the negative constant involved*:
>
> **My comment**: I read the comment and double-checked the proof, but still I cannot agree with the authors' rebuttal comment.
>
> First, if your comment were true, I would claim that the current draft has a significant presentation issue since the Theorem does not mention the importance of the constant. It even lacks an explanation about the sign of the constant. Moreover, even the explanation after Theorem 3 *Theorem 3 implies that the sum of the effective rank...* does not mention the importance of the constant.
>
> **In the first place**, according to equation (15) in the proof of Theorem 3, we only see constant $-\\log 4$ on the right-hand side. The constant 2 you mentioned in your answer does not appear since the positive one and negative one cancel out. As a result, by noting that the inequality in the proof reverses the sign in the Theorem, what the exact result looks like is $- D(\\mathbf{\\mu}, \\mathbf{\\Sigma}) \\le + \\log [\\mathrm{Erank} (\\mathbf{\\Sigma})] + \\log 4$. The left-hand side is non-positive and the right-hand side is non-negative plus positive, which is positive. Hence, the inequality is still trivial and does not contribute to the justification of the Erank.
>
> By the way, even if the constant were negative, I do not think it would justify Erank sufficiently. The left-hand side can go to negative infinity, and the right-hand side could not be smaller than the negative constant. The inequality would be too loose. Also, if the constant is, say, $\\log 4 - 2$, then the inequality is meaningful only when $D(\\mathbf{\\mu}, \\mathbf{\\Sigma}) \\le 2 - \\log 4$. It does not justify Erank well.
>
> - **2.** *This is not correct. The reviewer selectively highlights the case where Erank has attained its maximum, i.e, when the singular value $\\sigma$ of follows a uniform distribution.*
>
> **My comment**: First, even if the phenomenon happened only where Erank has attained its maximum, the authors should be responsible for explaining why it would not be a practical issue. What the author claimed is just that it is not an issue if the case I mentioned does not happen.
>
> **In the first place**, "*selectively highlights the case where Erank has attained its maximum*" is not a correct claim. I showed that case just to see the issue easily. Actually, for any $\\mathbf{\Sigma}$, we can see that $D(\\mathbf{\\mu}, c \\mathbf{\\Sigma})$ strongly depends on $c$, but $\\mathrm{Erank} (c \\mathbf{\\Sigma})$ does not depend on $c$.
>
> - **3.**
>
> **My comment**: Since the time is limited, I do not discuss it more, but I humbly suggest the author put ERank directly in the loss function in the future, if ERank represents the degree of dimensional collapse well.
>
> - **4.**
>
> **My comment**: I'm sorry to hear that my intention was not clear. What I wanted to say is that the disadvantage caused by the strong concentration effect caused by the KL divergence from the isotropic Gaussian distribution could surpass the benefit you receive from by forcing the learning results to be uniform by the KL divergence term.
>
> - **5.**
>
> **My comment**: Thank you for your answer. I'm aware that we limit the boundary in the Poincare disk model. I pointed it out as a presentation weakness of your paper. Here, I encourage the author to describe the **reason**, not who used them.
>
> - **6.**
>
> **My comment**: I'm sorry but I do not understand your answer here. If the topology of the graph has little thing to do, why did you draw a tree in Figure 4? Also, in which sense, why we could call it tree-embedding if the topology of the graph has little thing to do?
>
> - **7.**
>
> **My comment**: Your answer helps and I appreciate it. There is a reason to focus on this point. Other operations in your method seem $O(D)$, but only this determinant part is $O(D^{3})$ if we calculate it naively. Hence, it would be worth mentioning that this part's computation is not a practical issue.
>
> - **8.**
>
> **My comment**: $D$ exists after the first equal symbol. Regarding $p$, the author only says "two distributions $p$ and $q$," but I do not think Equation (9) holds for general $p$ and $q$. I guess $p$ must be the multivariate standard normal distribution, but it is not mentioned in the paper.
>
> - **10.** *We explained how Th. 1 matters to our work, not what original author's intentions were.*
>
> **My comment**: In the current manuscript, you wrote "*Such a property is formally discussed by Ganea et al. (2018) who **state** that any tree can be embedded into a Poincaré disk (n = 2) with low distortion*". So, the reader must think it is what original author's intentions were.

---

> ### Comment · Reviewer_WCaW · 2023-11-22
>
> - **10.** *We are happy to rephrase. A one minute job.*
>
> I do not think that it suffices to just rephrase your expression. It is because you used this theorem to justify your claim *the hyperbolic space is suitable for embedding hierarchical structures with constant branching factors and an exponential number of nodes.* However, Theorem 1 could justify using a hierarchical graph to approximate the hyperbolic space, but not its converse. Hence, it involves rewriting many parts of your paper.
>
> Perhaps you might have wanted to cite
> Sarkar, Rik. "Low distortion delaunay embedding of trees in hyperbolic plane." In International symposium on graph drawing, pp. 355-366. Berlin, Heidelberg: Springer Berlin Heidelberg, 2011.
>
> I believe this paper matches your intention. Hope it helps.

---

> > ### Comment · Reviewer_WCaW · 2023-11-23
> >
> > **Overall**, through the discussion with the authors, I have become confident that the current manuscript has many mathematical soundness issues. Unless these issues are essentially corrected, the acceptance of the current manuscript would be disadvantageous not only for the ICLR community but also for the authors' benefits. For this reason, I cannot raise my score. Also, I would like to increase my confidence from 4 to 5 since I have become confident that the issues I pointed out are not my misunderstandings through communicating with the authors. I strongly encourage the authors to inspect all the mathematics and their interpretations in the paper to convey their impressive ideas correctly to readers.

---

> ### Author Response · Authors · 2023-11-23
> **A very quick response on point 1.**
>
> **We apologize, we think reviewer is right here.** We should have checked point-by-point the proof of our junior colleague here. So thank you for picking this issue.
>
>
> Thank you for highlighting that term 2 is canceled out in Eq.15, and we acknowledge that the inequality scaling in Eq.16 might too aggressive and compromise the effective rank, resulting in a looser bound than intended for our claim.
>
> However, by looking at Eq. 15 without further inequality scaling in Eq.16, instead of bound on effective rank, one may use the bound on the condition number $(\frac{\lambda_1}{\lambda_d})$ as
> $$
> D(\boldsymbol{\Sigma}, \boldsymbol{\mu}) \geq\left(\log \frac{\lambda_1}{\lambda_d}\right)-\log4
> $$
>
> **This also implies that $D(\boldsymbol{\Sigma}, \boldsymbol{\mu})$ is trying to decrease the leading eignvalue $\lambda_1$ and increase the smallest eigenvalue $\lambda_d$, which make the spectrum to be uniform and increase overall entropy (effective rank).**
>
>
> Even without this earlier bound there seems to be quite close connection between eranks in the ambient space and the tangent space empirically, so they still are somewhat useful in terms of measuring dimensional collapse.
>
> In a way, all that we need to care about to **detect** dimensional collapse is erank in the ambient space as usually encoder is the one posing collapse. We used erank as simply this is used for dimensional collapse.
> \
> \
> **Re. point 2:** yes, the scaling effect of $c$ changes Eq. 9 because we do not scale ${\bf I}$ in Eq. 9. We will see if we introduce scale for  ${\bf I}$ in Eq. 9. Perhaps we will do this based on the maximum over norms of features in the tangent space. We think this issue can be resolved with minor modifications.
>
> Nonetheless, Eq. 9 ensures spectrum of covariance matrix to be close to that of identity matrix, then Erank and KL are then related. We will improve that argument but as network converges, we would expect Eq. 9 to reasonably impose that our normal distribution is not close to $c{\bf I}$ but ${\bf I}$. Would you not agree?
> \
> \
> In that sense, we felt reviewer's argument was cherry-picked and still see it a bit cherry-picked. **We assumed it is clear that if we impose Eq. 9 (through Eq. 10) as a part of our overall loss, then it would be obvious to the reader that we are not going to converge to $\mathcal{N}(0,c{\bf I})$ (say $\mathcal{N}(0, 2{\bf I})$ ) but rather closer to $\mathcal{N}(0,{\bf I})$.** Is that not obvious?
>
> **Re. point 6:** I believe there is some misunderstanding here. Are you talking about the topology of the input graph to the network?  Or are we talking about the topology of something else?
>
> **Re. point 7:** Justifying things based on $O(d)$ or $O(d^3)$ is not particularly telling in terms of actual speed. We would have to go into actual flops here but that is beyond the point of this paper (several others investigate applications of log det divergence in DL and times). Usually, the compute bottleneck is with backpropagating through the backbone not with computing the determinant. In fact, here we insist this as there are tons of practical deep learning works using logdet in such scenarios. Our own measured times barely differ whether we use logdet or not.
>
> **Re. point 8:** Ah, we see now which $D$ you referred to, yes, D is KL divergence between the Normal distribution (used for $q$) and the isotropic normal distribution (used for $p$), hence the identity matrix showing up in the equation 9. We will fix it accordingly.
>
> **Re. point 10:** Thank you. This work looks indeed interesting. Let us see how we can use it in our context.

---

> > ### Comment · Reviewer_WCaW · 2023-11-23
> >
> > - **Regarding point 1**:
> >
> >   I did not have time to check the proof of your new bound carefully, but let us assume the bound is correct (we may have room for improvement in the future, by the way). Now, your paper does not have a direct connection between your loss function and Erank. However, your manuscript discusses your results using Erank. This obviously needs a major revision. In the first place, you would need to explain why you did not directly include the Erank or $(\\log \\frac{\\lambda\_{1}}{\\lambda\_{d}})$ in your loss function but instead $D (\\mathbf{\\mu}, \\mathbf{\\Sigma})$.
> >
> > - **Regarding Re. 2**:
> >
> >   I do not see why the reviewer called it "cherry-picked". I stress here that the phenomenon happens for a **general** $\\mathbf{\\Sigma}$.
> >
> > - *We assumed it is clear...*:
> >
> >   The phenomenon itself would be relatively obvious, but readers would want to know why you did not try to exclude such a phenomenon (sensitivity to the scale of $\\mathbf{\\Sigma}$) since there is no clear description of the motivation for keeping the phenomenon in your loss function. This phenomenon is not directly related to your motivation, "uniformity" or avoiding "dimensional collapse".
> >
> > - **Regarding Re. 6**:
> >
> >   It was about the topology of the graph. Roughly speaking, a graph, including a tree, is a vertex set with topological information, including distance information, defined by the edges. If the topology were not important, we would have no motivation to discuss a graph. You could simply discuss a point set (without edges) and you should omit edges in the figure.
> >
> > - **Regarding Re. 7**:
> >
> >   I appreciate your answer and expect the discussion will be included in the manuscript in the future.
> >
> > - **Regarding Re. 8, 10**:
> >
> >   I appreciate the authors' answers.

---

> ### Author Response · Authors · 2023-11-23
> **thank you**
>
> Overall,
>
> We think we have mostly converged in the discussion with the reviewer. We thank for the scrutiny here and apologize for the issue with the bound (kindly see our response "A very quick response on point 1").
>
> We still think some arguments though are a bit cherry-picked as that about plugging covariance of $\mathcal{N}(0,c{\bf I})$ into Eq. 9 while Eq. 9 as a part of overall loss (through Eq. 10) encourages covariance of $\mathcal{N}(0,{\bf I})$ rather than $\mathcal{N}(0,c{\bf I})$ (e.g., we reasonably expect it be closer to $\mathcal{N}(0,{\bf I})$ than say $\mathcal{N}(0,2{\bf I})$ or $\mathcal{N}(0,100{\bf I})$  )- kindly see our updated response. To resolve that we can either rescale ${\bf I}$ in Eq. 9 or make assumptions that relation of ERank to KL holds upon convergence where Eq. 9 is assumed to closely enforce covariance close to identity.

---

> > ### Comment · Reviewer_WCaW · 2023-11-23
> >
> > Thank you for your comment. As in my previous reply, it is not cherry-picked at all since it holds for general $\\mathbf{\\Sigma}$.
> >
> > I know the ERank and KL have some relation. I did not say they are not related at all. What I criticized was that your manuscript did not explain the relation correctly, showing easy examples everyone can understand.

---

> ### Author Response · Authors · 2023-11-23
> **Regarding comment 4.**
>
> Dear Reviewer,
> \
> \
> One thing that would be useful for us to understand is what you mean by **the disadvantage caused by the strong concentration effect caused by the KL divergence from the isotropic Gaussian distribution could surpass the benefit you receive from by forcing the learning results to be uniform by the KL divergence term**.
> \
> \
> The way we see it is as follows. We applied the exp map transformation from the tangent space to the ambient space. We encourage isotropic Gaussian in the tangent space. The distribution of that isotropic Gaussian (after pushing it through exp map transformation) into the ambient space is in Figure 7.
> \
> \
> Thus, our Eq.9 and 10 promote such a distribution of features in the ambient space.
>
> If one samples the actual distribution:
> * the highest density is placed along the outer shell ring region (along the Poincare disc boundary)
> * the density along the ring (chose some suitable radius) is uniform and higher than on other rings formed closer towards the centre
> * the density towards centre of Poincare disk decreases (faster than linearly)
>
> This is precisely what we desire.
> \
> \
> So we are trying to understand what **the disadvantage caused by the strong concentration effect** is on your mind because obviously we are keen to address the issue but we kind of do not see what the issue exactly is. The only potential issue that we see is that **we may for example prefer density to decrease faster than currently is as one moves from the boundary of Poincare disc towards its centre.** Is that what the reviewer meant?
> \
> \
> Or reviewer means that KL on a covariance formed from features on the tangent space may have some unexpected effects when trying to align $\bf \Sigma$ with the identity matrix? (we guess rev. may be talking about this?)
>
> Best regards,
> \
> Authors

---

> ### Author Response · Authors · 2023-11-23
>
> * **Re. 1:** Including ERank on singular values in the actual code seems a tricky task, e.g., that generally means we would have to run SVD. Using condition number as well would generally require SVD. We use $D$ obtained from the KL divergence between $\bf \Sigma$ and $\bf I$ precisely because then the entire problem simplifies to a very basic logdet formulation, and logdet formulation enjoys a very friendly support in PyTorch compared to SVD (e.g., for SVD the moment one gets non-simple eigenvalues such as e.g. any two eigenvalues are equal one (or just are equal to each other in general), then the backpropagation through SVD is undefined so that is the first realistic challenge).
> \
> \
> Essentially, the moment we would use SVD to compute ERank and have some loss w.r.t. ERank, the reviewer's argument would be this is impractical because SVD is slow (and for sure much slower than logdet or logdet's approximation) and the code would simply throw non-convergent SVD due to some singular values being non-simple - and so the merry goes round - there is no way to win the argument, is there?
> \
> \
> Also, the issue is that prior works use ERank to measure the dimensional collapse so it is also not easy to replace ERank with other measure and convince all reviewers that is the way to go, as then of course we need to discuss at length why we did not use ERank, why other measure would be sufficient to prove that point, what are its properties...
>
> * **Re. 2**, general $\bf \Sigma$ would be OK and we understand the argument, but as D aligns  $\bf \Sigma$ with ${\bf I}$ in the loss via Eq. 9 and 10, that also has impact on controlling the scale and some unconstrained  $\bf \Sigma$ cannot happen. This is what we of course can explain, and analyze further. The point is, one should look at what the combined loss does to the scale, not just look at D alone because in that case you are right, there is the scale issue.
> \
> \
> Or we can simply look at ways to normalize the scale. We do not see how this is essential to strictly enforce scale invariance if we can encourage specific scale of  $\bf \Sigma$ as we push it towards  ${\bf I}$ via Eq. 9 and 10 plugged into the total loss.
> \
> \
> Of course, it is easy to see that all-ones spectrum enjoys a favourable ERank compared to say fast decaying spectrum clipped from top by 1.
> \
> \
> Either way, we absolutely can explain this in the draft, and analyze further the actual scale.
>
> * **Re. 6**, we are happy to rework that. We simply hint here at the connection of hyperbolic spaces to hierarchical representations - something every paper on hyperbolic DL talks about, be it Section 3.1 on the following paper or Eq. 11 of the second following paper.
> > Hyperbolic Image Embeddings, CVPR 2019
>
>   > Hyperbolic Entailment Cones for Learning Hierarchical Embeddings
>
>   We agree, Theorem 1 says that distance among points in hyperbolic space can be approximated by a tree subject to the right    mapping f. Indeed, it does not say the converse that any tree can be embedded in hyperbolic space.
>   \
>   \
>   To be absolutely honest, we are more than happy to remove the entire Theorem 1 as its removal changes nothing for the paper (as in, nothing in the paper relies on plugging it into Theorem 1).
>   \
>   \
>   When we talk about the dimensional collapse in the hyperbolic space, we know the $\ell_2$ norms of feature vectors in the ambient space  are related to the tree depth (of course that depth depends also on the negative curvature $c$). We know that the region within the shell (or ring to be more precise) in the ambient space should have uniform distribution as the collapse is something happening due to encoder being unable to fill certain parts of ambient space along the shell (to say it roughly). So, these are the properties we really care about here.
>   \
>   \
>   As such, they affect the tree depth, and its layout in the ambient space but we really do not want to go into the entire topology of the tree graph, as that is way beyond simple points we are trying to illustrate.

---

### Official Review · Reviewer_vDHc · 2023-10-27

**Soundness:** 2 fair
**Presentation:** 2 fair
**Contribution:** 3 good
**Rating:** 5
**Confidence:** 4

**Summary:**

Authors propose a novel graph contrastive learning framework called HyperGCL that relies on the projections of GNN node embeddings into hyperbolic spaces. For a given graph, two views are randomly produced by applying random perturbations (dropping egdes and nodes). These are fed to a 2-layer GCN to get node embeddings, which are then projected onto a Poincaré ball. A loss is then designed to achieve representation learning in this space. A first classical term based on the distance in this hyperbolic space is proposed to promote alignment of views. A second term handles the diversity of these representations by ensuring that embeddings populate the ball uniformly. To this end, authors propose to map embeddings into an Euclidean space where representations are forced to fit an isotropic gaussian distribution. This method allows to prevent the well-known dimensional collapse for which authors propose a clear taxonomy when learning in hyperbolic spaces.
Then they benchmark their method to several supervised GNN and contrastive methods using common evaluation in this literature, for node classification and collaborative filtering. HyperGCL consistently outperforms other compared methods across 8 datasets. Finally, the authors analyze both theoretically and empirically the effectiveness of HyperGCL in preventing dimensional collapse using the notion of effective rank.

**Strengths:**

-	Clear presentation of involved mathematical concepts
-	Clear analysis regarding the collapse modes when learning representations in hyperbolic spaces
-	Extensive benchmark of HyperGCL when learning representations in an unsupervised manner. Supervised evaluation of these representations show that HyperGCL is a novel SOTA contrastive graph learning method.
-	Pertinent analysis in terms of Effective rank with an interesting theoretical result.
-	Interesting ablation studies w.r.t the topology used to enforce alignment and uniformity.
-	Interesting analysis w.r.t the choice of gaussian distributions.

**Weaknesses:**

**1. On the form**: as a consequence few points are not so clear in substance.

  - a)	There are many tipos in the paper, please correct them.
  - b)	Figure 1 and 2 and their respective analysis in the text are not clear. Figures and their explanations could be improved / completed. As such, I would even suggest to move them in the supplementary material to have enough space to complete them.
  - c)	Definition 2 could be made clearer.
  - d)	Figure 4: modes of collapse should be put on the subplots. Clean tree should be isolated.
  - e)	Figure 5: in the text you say that you randomly drop edges and nodes. But in the figure it seems that you rather mask some node features not completely remove nodes.

**2. contextualizing research**
   - a)	I think that mappings from hyperbolic spaces to gaussian distributions were already studied in the Machine Learning literature e.g [A]. This should be clear in the paper. Also it seems to mitigate the theoretical contribution stated in Theorem 2.
   - b)	Optimization in Riemannian framework is clearly an active field of research whose advances seem to be disregarded by authors e.g [B], [C]. Could you further justify your choice for RSGD and perform simple benchmarks with other more recent approaches ?
   - c)	Various clearly competitive approaches are not benchmarked e.g GraphMAE [D] and more importantly CCA-SSG [E]. The latter also conducts a reflection to circumvent to dimensional collapse when operating in an Euclidean setting. The uniformity loss essentially comes down to enforcing embedding covariances to be close to an identity matrix like the one of an isotropic gaussian distribution. [E] also establishes relations between CCA-SSG and Mutual Information maximization under gaussian assumptions so in substance it clearly seems like an Euclidean counter to HyperGCL. As such clear comparisons should be present in the paper. Interestingly, It seems that these methods need a higher embedding dimension to compete/outperform HyperGCL which was benchmarked for a fixed embedding dimension of 50 but where the curvature parameter needs to be fine-tuned.

**3. Incomplete experiments or analysis.**
   - a)	What are the hyperparameters involved in the perturbation strategy ? Is there a validation of these parameters and if so what is the sensitivity of the method to these hyperparameters ?
   - b)	Could you provide the sensitivity analysis to the curvature parameter on the node classification part.
   - c)	No experiments in semi-supervised learning settings. As HyperGCL seems to provide discriminant embeddings keeping low-dimensional embeddings, I would tend to believe that they would better suit semi-supervised learning than Euclidean contrastive graph learning method. As CCA-SSG seems on par with HyperGCL (see 2.c)) , it is not obvious that the overall hyperbolic setting is better than the Euclidean one.
   - d)	Lack of clarity or hindsight w.r.t the evaluation : No clear justifications for the choice of supervised evaluation. No fully unsupervised evaluations proposed which would suit the learned topology.
   - e)	No sensitivity analysis w.r.t the encoder. Nor a clear comparison between performances of this GNN backbone in a fully supervised setting vs the 2-step strategy used by authors to evaluate HyperGCL embeddings. Such analysis could relate to the common concerns in the GNN literature e.g i) expressivity simply considering e.g several GNN layers using Jumping Knowledge based backbones; ii) homophily vs heterophily via e.g [F] whose supervised models exhibit considerably higher classification performances than those reported in Table 1.

[A] Mathieu, E., Le Lan, C., Maddison, C. J., Tomioka, R., & Teh, Y. W. (2019). Continuous hierarchical representations with poincaré variational auto-encoders. Advances in neural information processing systems, 32.

[B] Becigneul, G., & Ganea, O. E. (2018, September). Riemannian Adaptive Optimization Methods. In International Conference on Learning Representations.

[C] Kasai, H., Sato, H., & Mishra, B. (2018, July). Riemannian stochastic recursive gradient algorithm. In International Conference on Machine Learning (pp. 2516-2524). PMLR.

[D] Hou, Zhenyu, et al. "Graphmae: Self-supervised masked graph autoencoders." Proceedings of the 28th ACM SIGKDD Conference on Knowledge Discovery and Data Mining. 2022.

[E] Zhang, Hengrui, et al. "From canonical correlation analysis to self-supervised graph neural networks." Advances in Neural Information Processing Systems 34 (2021): 76-89.

[F] Luan, Sitao, et al. "Revisiting heterophily for graph neural networks." Advances in neural information processing systems35 (2022): 1362-1375.

**Questions:**

I invite the authors to discuss the weaknesses I have mentioned above and to provide additional results/analyses for refutation, knowing I am inclined to increase my score. Follows some questions to clarify some points:

Q1. Do you normalize embeddings or just center them ?

Q2. The choice of  a linear classifier to evaluate clearly non-linear embeddings is not so obvious  (I know that it is not questioned in most graph contrastive learning papers). Could you provide complementary evaluations with a non-linear classifier e.g 2-MLP with ReLU activation ?

---

> ### Author Response · Authors · 2023-11-22
> **Rebuttal (Rev. vDHc) - part I**
>
> # 1.1. There are many tipos in the paper, please correct them.
>
> Thank you. Of course, we have now had another iteration of proofreading. We will fix all typos.
>
> # 1.2 Figure 1 and 2 and their respective analysis in the text are not clear.  I would even suggest to move them in the supplementary material to have enough space to complete them.
>
> Duly noted. Limited space was exactly the reason why these \`dimensional collapse\' cases are not described better by us.
>
> # 1.3. Improve Definition 2.
>
> Thank you. We simply followed here standard literature here (not our contribution). Nonetheless, we will improve it accordingly.
>
> # 1.4.  Figure 4: modes of collapse should be put on the subplots. Clean tree should be isolated.
>
> Duly noted.
>
> # 1.5. Figure 5: in the text you say that you randomly drop edges and nodes. But in the figure it seems that you rather mask some node features not completely remove nodes.
>
> Thank you. Indeed, we use edge drop and attribute mask just as the majority of other methods do. We have now corrected the text.
>
>
> # 2.1. Mappings from hyperbolic spaces to Gaussian distributions studied in [A].
>
> Thank you. That is a very interesting paper. It appears our expression differs from that in [A]. Nonetheless, there is a connection and we will gladly acknowledge it.
>
> # 2.2.  Optimization in Riemannian framework is clearly an active field of research [B,C] whose advances seem to be disregarded by authors.
>
> Kindly notice that **the optimization on Riemannian manifolds is not the topic of our paper**, at all. We simply use RSGD just as a baseline optimization tool, just as the paper [B] does use RSGD as a baseline. Nonetheless, it is a very interesting pointer as RADAM works a little bit better with a quicker convergence. Table below presents our current results:
>
>
> |Method  |Disease         | Airport         | PubMed          |CiteSeer         |Cora            |
> |---|---|---|---|---|---|
> |HyperGCL+RSGD| 94.50 $\pm$0.43  |   93.55 $\pm$1.11 |  85.14$\pm$ 0.38  |  73.43 $\pm$ 0.35  |  84.47 $\pm$ 0.46 |
> |HyperGCL+RADAM| **94.61 $\pm$ 0.51**  |   **93.68 $\pm$ 0.97** |  **85.40 $\pm$ 0.25**  |  **73.68 $\pm$ 0.22**  |  **84.77 $\pm$0.62** |
>
> # 2.3.  Various clearly competitive approaches are not benchmarked e.g GraphMAE [D].
>
> Thank you. We are more than happy to add GraphMAE to discussions and comparisons. However, **GraphMAE is a Masked Auto-Encoder (MAE) rather than a Contrastive Learning (CL) approach**.
> \
> \
> Currently, in self-supervised learning, **two families (masked AE and CL) are used as complementary techniques** as they all exhibit very different strengths, i.e., different behavior for linear probing vs. fine-tuning etc.) In fact, they even produce entirely different characteristics of embedding spaces, as shown in several studies.
> \
> \
> A fair way would be not to compare our method with GraphMAE, but in fact combine GraphMAE with our CL, as in current works that combine both MAE and CL. However, this is clearly beyond the scope of our work, as that alone could be a separate paper. Thus, we are not devising mixed Graph MAE+CL methods as in the LGP paper:
> > Layer grafted pre-training: Bridging contrastive learning and masked image modeling for label-efficient representations, ICLR 2023
>
>
> Thank you, **in comparisons with CCA-SSG [E] our approach appears better**:
>
> | | Cora | Citeseer | Pubmed |
> |-|-|-|-|
> |CCA-SSG [E] |  84.2 |  73.1 | 81.6  |
> |HyperGCL+RSGD  |  **84.47**   |  **73.43** |  **85.14** |
> |HyperGCL+RADAM|  **84.77** |  **73.68**  |  **85.40** |
>
> # 2.3b. The uniformity loss essentially comes down to enforcing embedding covariances to be close to an identity matrix like the one of an isotropic gaussian distribution. Euclidean counter to HyperGCL.
>
> Kindly notice we compared in our paper with the Euclidean counterpart of HyperGCL too in Table 3 (the first row of table imposes uniformity). They all performed worse just as CCA-SSG performs worse.
>
> In fact, **we observed that typical soft-uniformity on covariances still suffers from the dimensional collapse  as there is nothing to ensure the infinite penalty when an eigenvalue of covariance collapses to zero.** That is, if other loss components benefit at the cost paid by the soft-penalty, in CCA-SSG the collapse will happen.
>
> **That is why CCA-SSG performs worse than our approach that enjoys logdet penalty.**(our logdet sets an infinite penalty under  eigenvalue of covariance going to zero - thus, collapse as in nullifying an eigenvalue under logdet is too costly to occur).

---

> ### Author Response · Authors · 2023-11-22
> **Rebuttal (Rev. vDHc) - part II**
>
> # 3.1 What are the hyperparameters involved in the perturbation strategy? Is there a validation of these parameters and if so what is the sensitivity of the method to these hyperparameters?
>
> The choice of perturbation hyperparameters is heuristic, we grid-search on the hyperparameters (edge drop ratio and attribute mask ratio) on the validation set. Table below presents selected hyperparameters:
>
> |Disease |Airport |Cora |PubMed |CiteSeer|
> |---|---|---|---|---|
> |Edge drop ratio        |0.1      |0.2        |0.2       |0.1          |0.3|
> |Attribute mask ratio   |0.0      |0.0        |0.4       |0.3          |0.2|
>
>
>
> # 3.2 Could you provide the sensitivity analysis to the curvature parameter on the node classification part.
>
>
>
> |c |Disease |Airport|
> |---|---|---|
> |c = 0.05  |92.74$\pm$0.22 |92.15  $\pm$ 1.31|
> |c = 0.1  |94.20$\pm$0.26 |93.53  $\pm$ 1.10|
> |c = 0.2  |93.60$\pm$0.14 |91.98  $\pm$ 1.13|
> |c = 0.3    |92.80$\pm$0.16 |91.02  $\pm$ 1.02|
>
>
>
> # 3.3 No experiments in semi-supervised learning settings. As HyperGCL seems to provide discriminant embeddings keeping low-dimensional embeddings, they would better suit semi-supervised learning than Euclidean contrastive graph learning method. As CCA-SSG seems on par with HyperGCL (see 2.c)) , it is not obvious that the overall hyperbolic setting is better than the Euclidean one.
>
> Thank you. In fact, CCA-SSG performs much lower on Disease and  Airport than our HyperGCL:
>
> |Disease | Airport |Cora |PubMed |CiteSeer|
> |---|---|---|---|---|
> |CCA-SSG   |69.21$\pm$0.3 |81.98$\pm$0.4  |  84.2$\pm$0.4 |73.1$\pm$0.3 | 81.6 $\pm$0.4|
> |GraphMAE  |70.61$\pm$0.5  |82.79$\pm$0.4 |  84.2$\pm$0.4 |73.2$\pm$0.4 | 81.1 $\pm$0.4|
> |HyperGCL  |**94.50 $\pm$0.4** |**93.55$\pm$1.1** |  **84.4 $\pm$0.4** |**73.4 $\pm$ 0.3** | **85.1 $\pm$ 0.3**|
>
>
> # 3.4 Lack of clarity or hindsight w.r.t. the evaluation: No clear justifications for the choice of supervised evaluation. No fully unsupervised evaluations proposed which would suit the learned topology.
>
> Supervised evaluation follows the typical GNN evaluation setting described in many papers, e.g.:
> > Semi-Supervised Classification with Graph Convolutional Networks ICLR 2016
>
> The self-supervised setting follows:
> > COSTA: Covariance-Preserving Feature Space Augmentation for Graph Contrastive Learning KDD 2022
>
> # 3.5 No sensitivity analysis w.r.t the encoder. Nor a clear comparison between performances of this GNN backbone in a fully supervised setting vs the 2-step strategy used by authors to evaluate HyperGCL embeddings. Such analysis could relate to the common concerns in the GNN literature e.g i) expressivity simply considering e.g several GNN layers using Jumping Knowledge based backbones; ii) homophily vs heterophily via e.g [F] whose supervised models exhibit considerably higher classification performances than those reported in Table 1.
>
> Since we could not find the code to reproduce the result in [F],  we compare different GNN backbones (e.g., GCN, GAT, JKNet) on Disease/Airport with supervised setting and SSL setting (2-step) with HyperGCL:
>
>
> |Method |Disease (Supervised) | Airport (Supervised) | Disease (SSL HyperGCL(2-step)) | Airport (SSL HyperGCL(2-step)) |
> |-|-|-|-|-|
> |GCN   |69.61$\pm$0.5 |81.40 $\pm$ 0.4 |  94.5 $\pm$0.4| 93.57$\pm$1.1|
> |GAT   |70.40$\pm$0.4 |81.50  $\pm$ 0.3 |  93.8 $\pm$0.3| 92.72$\pm$1.2|
> |JKnet |68.80$\pm$0.4 |80.53  $\pm$ 0.3 |  92.6 $\pm$0.4| 92.44$\pm$0.9|
>
>
> # 4.1 Do you normalize embeddings or just center them?
> We do not normalize embeddings, we only center them.
>
>
> # 4.2 The choice of a linear classifier to evaluate clearly non-linear embeddings is not so obvious (I know that it is not questioned in most graph contrastive learning papers). Could you provide complementary evaluations with a non-linear classifier e.g 2-MLP with ReLU activation?
>
> | |Disease | Airport |Cora |PubMed |CiteSeer|
> |---|---|---|---|---|--|
> Linear classifier         |94.50 $\pm$0.4 |93.55$\pm$1.1 |  84.4 $\pm$0.4 |73.4 $\pm$ 0.3 | 85.1 $\pm$ 0.3 |
> 2-MLP with ReLU   |94.82 $\pm$0.5 |93.98$\pm$1.4 |  84.3 $\pm$0.5 |73.8 $\pm$ 0.2 | 85.3 $\pm$ 0.5 |
>
> In general, there is some benefit to a non-linear classifier. Although in high-dimensional feature space features are mostly linearly separable. For Cora, there is even a drop in performance on 2-MLP with ReLU which we suspect may be a sign of overfitting.

---

> ### Comment · Reviewer_vDHc · 2023-11-22
> **Answer to reviewer**
>
> Thank you for these detailed answers.
>
> I strongly encourage the authors to implement in the paper the main comments I made on the form in my initial review.
> I find clearly interesting all additional results provided by the author during this rebuttal. Hence I also strongly encourage them to add these results and adequate discussions to the main paper (additional baselines + results RADAM for HyperGCL), or the supplementary material with adequate reference in the main paper (Semi-supervised evaluation, which clearly emphasizes the benefit of low-dimensional hyperbolic embeddings for generalization + sensitivity analysis). For the sake of reproducibility, I also suggest the authors to add an exhaustive table with the validation grids for HyperGCL, plus best hyper-parameters for each dataset.
>
> Overall I find authors' rebuttal really compelling. Assuming that my suggestions will be implemented in the paper, I will increase my score from 5 to 7 (Accept).

---

> > ### Author Response · Authors · 2023-11-22
> > **Thank you.**
> >
> > Thank you,
> >
> > Of course, we are more than happy to include these new observations and results into the paper as per our discussions. We will start work on a revision accordingly. We really appreciate this good quality feedback that helps us refine the paper with many additional points.
> >
> > Meantime, if there is anything else we can answer or improve, kindly let us know.
> >
> > Authors

---

### Official Review · Reviewer_NnrR · 2023-10-28

**Soundness:** 3 good
**Presentation:** 3 good
**Contribution:** 2 fair
**Rating:** 5
**Confidence:** 3

**Summary:**

The paper tackles the challenge of learning effective self-supervised graph representations, with a focus on hyperbolic spaces. While contrastive learning is effective in Euclidean spaces, graphs often have non-Euclidean latent geometry. The proposed contrastive learning framework introduces an alignment metric capturing hierarchical data-invariant information and addresses the dimensional collapse issue. The authors associate dimensional collapse with "leaf collapse" and "height collapse" in hyperbolic spaces, proposing a graph contrastive learning framework that operates in the hyperbolic space. To mitigate dimensional collapse, they introduce an isotropic Gaussian loss on the tangent space of the hyperbolic manifold, promoting an isotropic feature distribution. The contributions include investigating the dimension collapse problem, proposing a hyperbolic graph contrastive learning framework, and introducing an isotropic Gaussian loss to address dimensional collapse.

**Strengths:**

- The paper is well-written.
- The proposed contrastive learning framework in hyperbolic spaces is interesting.
- The paper's emphasis on theoretical properties adds a robust theoretical foundation to the proposed framework.
- The framework's adaptability to various downstream tasks is highlighted.
- Detailed experimental information in the appendix enhances transparency.

**Weaknesses:**

- The paper's weakness lies in the observed marginal improvement in performance over prior methods. While the idea is intriguing, the practical impact of the proposed framework in terms of tangible performance gains may be limited.
- Personally, the empirical results do not convincingly demonstrate the advantages derived from the theorems.
- The absence of clear computational complexity advantages over existing methods is a point of consideration. Without distinct efficiency benefits, it becomes challenging to justify the adoption of the proposed approach, especially if it offers similar performance to existing methods. Addressing this aspect would enhance the paper's practical appeal.

**Questions:**

Please see the remarks mentioned above.

---

> ### Author Response · Authors · 2023-11-22
> **Rebuttal (Rev. NnrR)**
>
> # 1. The paper's weakness lies in the observed marginal improvement in performance over prior methods. While the idea is intriguing, the practical impact of the proposed framework in terms of tangible performance gains may be limited.
>
> Thank you. We do not consider the improvements to be marginal, given that the model has achieved State-of-the-Art (SOTA) performance across six graph datasets and three recommendation datasets.
>
> **While datasets may display indicators nearing saturation** (broader problem in graph learning), suggesting limited potential for large improvement gains on their own, our method demonstrates the ability to continually enhance performance.
> \
> \
> This constantly present improvement underscores the superiority of our approach, even in scenarios where immediate gains might seem difficult for other methods to achieve.
>
> |Method  |Disease         | Airport         | PubMed          |CiteSeer         |Cora            |
> |---|---|---|---|---|---|
> |COSTA   | 67.12 $\pm$ 0.39  |   81.19 $\pm$ 0.40 |  84.31 $\pm$ 0.37  |  70.77 $\pm$0.24  |  82.14 $\pm$0.62 |
> |HGCL    | 93.42 $\pm$ 0.82  |   92.35 $\pm$1.51 |  83.14 $\pm$0.58  |  72.11 $\pm$0.64  |  82.37 $\pm$ 0.47 |
> |HyperGCL| 94.50 $\pm$ 0.43  |   93.55 $\pm$ 1.11 |  85.14 $\pm$0.38  |  73.43$\pm$ 0.35  |  84.47 $\pm$0.46 |
>
> Kindly notice **in all cases we have between 1-2\% gain demonstrating the main point of our easy-to-compute add-on.**
> \
> \
> **We have run also the statistical t-test** given we have deviation and numbers of runs. Except for Airport, in all cases our gains are **statistically significant** within 95\% confidence intervals.
>
> # 2. The absence of clear computational complexity advantages over existing methods is a point of consideration.
>
> The only overhead introduced in our method is to compute determinant. However, the determinant computation can be efficiently achieved with the Cholesky factorization even with larger feature dimensionality $d$.
> \
> \
> The following table show the time in ms w.r.t to dimension.
>
> d   | time(ms)|
> --- | ---  |
> 5   | 0.030|
> 10  | 0.040|
> 20  | 0.061|
> 40  | 0.123|
> 80  | 0.393|
> 200 | 2.223|
> 500 | 22.18|
> 1000| 119.7|
>
> In contrast to the naive approach in Eq. 6, such as calculating the pair-wise Riemannian distance between all point combinations, our method enjoys less computation.
> \
> \
> As can be seen, **our method just ads few milliseconds per sample which is negligible compared with the cost of running the backbone.**
>
> Cholesky has in practical terms complexity lower than the complexity of SVD $\mathcal{O}(\min(d^2 n, n^2 d))$. Kindly notice matrix determinant has very efficient parallelized implementation in PyTorch and is frequently used in deep learning papers:
> > Learning Log-Determinant Divergences for Positive Definite Matrices (TPAMI 2021)

---

> > ### Comment · Reviewer_NnrR · 2023-11-23
> >
> > I appreciate your attention to my concerns. After reading the other reviewers' comments and your rebuttals, I have decided to maintain my current score.

---

> > > ### Author Response · Authors · 2023-11-23
> > > **thank you**
> > >
> > > Thank you,
> > >
> > > We are a little disappointed here because:
> > > * results, as in gains, are statistically significant (and we do not know of other works showing better results in our setting)
> > > * the speed due to log det is not an issie at ałl, as demonstrated.
> > >
> > > As such, have we not resolved reviewr's comments satisfactorily?

---

### Official Review · Reviewer_M3pp · 2023-10-29

**Soundness:** 3 good
**Presentation:** 3 good
**Contribution:** 3 good
**Rating:** 6
**Confidence:** 2

**Summary:**

In the paper "Alignment and outer shell isotropy for hyperbolic graph contrastive learning", the authors suggest a novel approach to hyperbolic (graph) contrastive learning. In contrastive learning, similar pairs of objects attract ("alignment" part of the loss function) while all pairs repulse ("uniformity" part of the loss). In usual contrastive learning, the embedding space is compact (hypersphere) and the uniformity loss aims to spread the embedding vectors uniformly. In contrast, hyperbolic embedding space has infinite volume, so uniformity loss would not make sense. The authors suggest an alternative, and apply this setup to graph contrastive learning. They show that the resulting GraphGCL outperforms all competitors on common benchmarks.

**Strengths:**

I am not familiar with the literature on hyperbolic graph neural networks, so can not really judge the novelty aspect. That said, I found the paper interesting: it suggests a new idea and shows that the resulting method outperforms existing methods. That hyperbolic embeddings would perform well for graph data, seems to make sense.

**Weaknesses:**

I do not have major criticisms. The suggested algorithm, HyperGCL, shows a marginal (~1 percentage point) improvement on all datasets that the authors analyzed.

**Questions:**

MAJOR COMMENTS

* The results tables (Table 1/2) look a bit "too good to be true": HyperGCL take the first place for every single dataset. Did the authors obtain all the values themselves (by running all the algorithms on all datasets)? Or are the values for other algorithms taken from the respective publications? This should be clarified.

* According to Table 1, HGCL is also a hyperbolic graph contrastive learning method. The authors should explain how it is different from HyperGCL. Is it the only hyperbolic graph contrastive learning method in existence?


MINOR COMMENTS

* page 3, Definition 2: should \mathbb D^n_c be D^d_c? Previously you only used D^d_c.

---

> ### Author Response · Authors · 2023-11-22
> **Rebuttal (Rev. M3pp)**
>
> # 1. Results of Table 1/2 look a bit "too good to be true": HyperGCL take the first place for every single dataset. Did the authors obtain all the values themselves (by running all the algorithms on all datasets)? Or are the values for other algorithms taken from the respective publications? This should be clarified.
>
> Thank you. These are the results we obtained. If they are good, we are glad.
>
> In Table 1, the results have been replicated using their official code.
> \
> In Table 2, the results are sourced from the HRCF paper.
>
> # 2. The authors should explain how HGCL is different from HyperGCL.
>
> * HGCL treats node embedding in the hyperbolic space as one view and the node embedding in the Euclidean space as another. It employs typical contrastive learning objective on these two views to learn node embeddings.
>
> * Our method utilizes both views in the hyperbolic space and aims to mitigate the dimension collapse through alignment and outer shell isotropy, which yield a good improvement.
>
>
> # 3. Should $\mathbb D^n\_c$ be $\mathbb  D^d\_c? $
>
> Indeed, thank you for noticing this. We have now fixed accordingly.

---

> > ### Comment · Reviewer_M3pp · 2023-11-22
> >
> > Thank you for you response. I have read the other reviews and your rebuttals and am keeping my positive score (6).

---

### Author Response · Authors · 2023-11-22
**Discussions**

Esteemed Reviewers,
\
\
**We thank for your valuable time and comments helping improve our work.**

**Although time is limited, we are here open to discussions with you.**
\
\
We hope our rebuttal clarifies all issues.

With regards,
\
Authors

---

### Meta-Review · Area_Chair_gXga · 2023-12-09

**Metareview:**

This paper proposes a contrastive learning method on the hyperbolic manifold. For theoretical contributions, the authors attempted to motivate the use of a KL-divergence term in the proposed formulation by bounding this term by a function of the effective rank of the covariance matrix. However, one of the reviewers pointed out a major technical issue regarding the correctness of the main theorem.

**Justification For Why Not Higher Score:**

The theory of proposed method has not been established rigorous enough for publication.

For the authors information, during the post-rebuttal period, Reviewer vDHc, despite hinting to increase the score from 5 to 7, eventually decided to keep the score at 5 and advocate for rejecting the paper, saying "Reviewer WCaW actually spotted a red-flag regarding Theorem 3 that I missed as I misread the exponential in the Erank definition. Of course the paper has merits on the practical side, but as Erank is at the core of authors' analysis they should definitely fix this. Therefore I will not change my score and advocate for rejecting the paper as reviewer WCaW suggests."

**Justification For Why Not Lower Score:**

N/A

---

### Decision · Program_Chairs · 2024-01-16

Reject